# Hippocampal-like Sequential Editing for Continual Knowledge Updates in Large Language Models

**Quntian Fang**[1*]    **Zhen Huang**[1*]    **Zhiliang Tian**[1†]    **Minghao Hu**[4†]    **Dongsheng Li**[1]
**Yiping Yao**[2]    **Xinyue Fang**[1]    **Menglong Lu**[3]    **Guotong Geng**[4]
[1]National Key Laboratory of Parallel and Distributed Computing
[2]College of System Engineering
[3]Key Laboratory of Advanced Microprocessor Chips and Systems
National University of Defense Technology
[4]Center of Information Research, AMS
{fangquntian,huangzhen,tianzhiliang,dsli,ypyao
fangxinyue,lumenglong}@nudt.edu.cn
{humh573,ggtong}@163.com

## Abstract

Large language models (LLMs) are now pivotal in real-world applications. Model editing has emerged as a promising paradigm for efficiently modifying LLMs without full retraining. However, current editing approaches face significant limitations due to parameter drift, which stems from inconsistencies between newly edited knowledge and the model's existing knowledge. In sequential editing scenarios, cumulative drifts progressively lead to model collapse characterized by general capability degradation and balance between acquiring new knowledge and catastrophic forgetting of existing knowledge. Drawing inspiration from the hippocampal trisynaptic circuit for continual memorizing and forgetting, we propose a Hippocampal-like Sequential Editing (HSE) framework that designs the unlearning of obsolete knowledge, domain-specific knowledge update separation and replay for edited knowledge. Specifically, the HSE framework designs three core mechanisms: (1) Machine unlearning selectively erases outdated knowledge to facilitate integration of new information, (2) Fisher information matrix-guided parameter updates prevents cross-domain knowledge interference, and (3) Parameter replay consolidates long-term editing memory through lightweight and global replay of editing data in a parametric form. Theoretical analysis demonstrates that HSE achieves smaller generalization error bounds, more stable convergence and higher computational efficiency. Experimental results validate its effective balance between acquiring new knowledge and mitigating catastrophic forgetting, maintaining or even slightly enhancing general capabilities. In practical applications, experiments confirm its effectiveness in multi-domain hallucination mitigation, healthcare knowledge injecting, and societal bias reduction. Our code is available at HSE_code [3]

## 1 Introduction

In real-world applications, large language models require frequent updates to correct erroneous or outdated knowledge [11, 54, 38]. Nevertheless, directly retraining LLMs incurs substantial

---

*contributed equally to this work.

†Corresponding authors.

[3]https://github.com/Square-Group-Sky/HSE

computational costs. Consequently, model editing has emerged as a paradigm to precisely modify LLMs' behavior for specific knowledge, while preserving performance on unrelated knowledge [7, 65, 67]. For practical deployment [49, 36], LLMs must integrate cutting-edge research while simultaneously discarding outdated knowledge and preserving validated knowledge. Therefore, sequential editing of LLMs further elevates the model editing approach to a continual learning paradigm, aiming to ensure that LLMs retain all edited knowledge across multiple editing operations and preserve the general capabilities [65, 67, 35]. Mainstream model editing methods focus on learning from static datasets, whereas they are less effective in handling sequential data streams [65, 33]. This poses significant challenges, particularly in dynamic environments where models need to continuously update and adapt to new knowledge.

To address the challenges of sequential editing, researchers have proposed two main approaches. (1) Parameter-preserving approach introduces additional modules integrated into LLMs, incorporating extra trainable parameters [22, 32] or network memory components [20] to record past edits while freezing the original model parameters. Nonetheless, this method tends to conservatively preserve the integrity of the original model parameters, which may limit its ability to adequately capture evolving data distributions. (2) Parameter-modifying approach focuses on modifying the model parameters directly to adapt to new knowledge. This approach first identifies the relevant parameters associated with the new knowledge and then computes an update matrix to modify [44, 14]. In comparison, the parameter-modifying approach enhances adaptation to new tasks by updating only the parameters directly associated with the task, leading to a more lightweight update process and superior generalization performance [65, 10]. Nevertheless, the parameter-modifying method may introduce conflicts between the LLMs' existing knowledge and the external editing knowledge [34]. In sequential editing scenarios, these conflicts can accumulate, leading to parameter drift in the edited layers of LLMs, making them incompatible with other parameter layers [15]. The accumulation of such drifts can subsequently lead to model collapse. Therefore, under sequential editing settings, the LLMs are susceptible to several issues: degradation of performance on general capabilities, and a delicate balance between acquiring new knowledge and catastrophic forgetting of existing knowledge. These issues make it difficult to apply existing model editing methods practically in sequential editing.

Fortunately, these issues are effectively addressed in biological systems, which exhibit continual learning and adaptation throughout their lifetimes [59, 18, 66]. The hippocampus, a key brain structure, possesses effective mechanisms for continual memorizing and forgetting [55, 51]. Synaptic plasticity in the hippocampus modulates neuronal activity by regulating synaptic strength, thereby facilitating both memorizing and forgetting [51, 2, 19]. During the processes of memory encoding, the hippocampus employs pattern separation to effectively distinguish between different input patterns [6, 17, 42], thereby enhancing the stability of memories. The replay mechanism that occurs during rest periods supports the transformation from short-term to long-term memory within the hippocampus [26, 52]. This process involves the reactivation of neural traces of previously experienced events, which significantly strengthen memory consolidation. Thus, the biological mechanisms within the hippocampus are essential for continual memorizing and forgetting processes, serving as an inspiration for the development of sequential editing.

In this paper, we propose a **H**ippocampal-like **S**equential **E**diting (HSE) method, which designs the unlearning of obsolete knowledge, domain-specific knowledge update separation and replay for edited knowledge. HSE designs the following editing strategies inspired by the biological mechanisms observed in the part of hippocampal trisynaptic circuit (DG→CA3→CA1) [30, 5]. (1) To address the conflicts between existing knowledge and editing knowledge, HSE adopts a memory-directed active forgetting strategy for machine unlearning to discard knowledge within LLMs that is inconsistent with the edits, inspired by how the hippocampus employs long-term depression (LTD) in CA3→CA1 to selectively forget outdated information [2]. (2) Given that new knowledge may originate from diverse domains, HSE utilizes a weight matrix (Fisher information matrix [48, 29]) to determine the importance of each parameter for the editing knowledge from different domains and appropriately controls update magnitude during sequential editing. This domain-separated mechanism is inspired by the way that dentate gyrus (DG) in hippocampus employs pattern separation mechanism to distinguish knowledge from different domains, thereby reducing interference between them [42]. This facilitates LLMs to edit knowledge from diverse domains while significantly reducing mutual interference. (3) To balance between acquiring new knowledge and catastrophic forgetting of existing knowledge, our proposed HSE derives a closed-form solution to promote the long-term editing memory of the LLMs, inspired by that hippocampus employs memory consolidation in the CA3 and CA1 region [26].

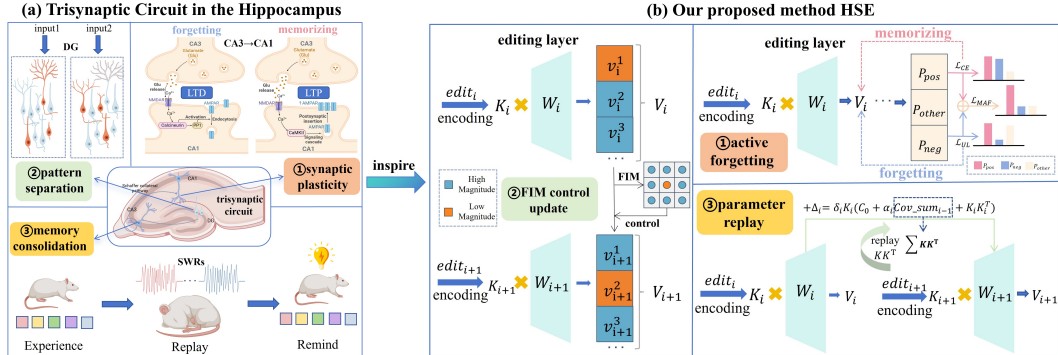

Figure 1: **Illustration of the HSE method inspired by the hippocampal trisynaptic circuit.** **(a)** The trisynaptic circuit within the hippocampus (DG→CA3→CA1), where ① LTP and LTD occurring between CA3 and CA1 are responsible for mechanisms of memory formation and forgetting, respectively. ② The DG handles pattern separation and ③ CA3 facilitates sharp-wave ripples (SWRs) in CA1 to consolidate memories. **(b)** Our proposed method HSE is inspired by biological mechanisms, including ① active forgetting based on machine unlearning, ② control of model editing parameter updates using the Fisher information matrix (FIM), and ③ long-term editing memory that reinforces edited knowledge and prevents parameter surge. $v_i^j$ denote the $j_{th}$ dimension of input and output for the $i_{th}$ editing knowledge, respectively.

Building on this biological mechanism, HSE progressively replays long-term editing memories to ensure the sustainability and convergence of the editing process.

Our experimental results show that HSE significantly outperforms existing model editing methods across multiple benchmarks. Compared to the best baseline, our approach demonstrates average improvements of 20.6% in generalization, 21.9% in specificity and 17.3% in efficacy. In contrast to other editing methods, which suffer a catastrophic drop in performance to zero, our approach not only preserves but also even improves the general capabilities of the original LLMs after sequential editing on ZsRE question answering datasets. Theoretical analysis confirms that the HSE method exhibits tighter generalization error bounds, more stable convergence and higher computational efficiency. Furthermore, HSE facilitates significant practical advancements in mitigating multi-domain hallucinations, updating healthcare knowledge and reducing societal biases for LLMs.

## 2 Methods

### 2.1 Task Formulation

For the $i_{th}$ editing operation on a LLM $f$, given the $i_{th}$ new knowledge (subject $s_i$, relation $r_i$, object $o_i^*$) with its corresponding old knowledge $(s_i, r_i, o_i)$, sequential editing task of LLM aims to integrate new editing knowledge while preserving previous edited knowledge and maintaining general abilities. The $i_{th}$ editing operation on $f$ results in $f_i$. The editing data is defined as $e_i = (s_i, r_i, o_i, o_i^*)$, and the editing operation is formulated as $f_{i+1} = \mathcal{E}_i(f_i, e_i)$. After applying a sequence of editing operations $E = \{\mathcal{E}_0, \cdots, \mathcal{E}_n\}$, the LLM evolves into $f_n$. For all editing data $\{e_i = (s_i, r_i, o_i, o_i^*) | i \leq n\}$, the LLM should generate the edited target objects $f_n(s_i, r_i) = o_i^*$. For data points $\bar{x}$ not associated with the editing data, the LLM should retain the original predictions, ensuring that $f_n(\bar{x}) \approx f_0(\bar{x})$, where $f_0$ denotes the initial LLM.

The second linear layer $W$ of the feed-forward network (FFN) in the early transformer modules is often regarded as a key-value associative memory [44, 43, 12]. This memory encodes the editing knowledge to ensure that the encoded input are accurately mapped to the target outputs. Due to the significant semantic information carried by the last subject token [43], the input encoding of the last subject token before entering edited linear parameter $W$ is denoted as $k \in \mathbb{R}^q$, and the output encoding of the updated object $o^*$ in $W$ is denoted as $v \in \mathbb{R}^d$. For a series of key-value knowledge pairs $K = \{k_1, \cdots, k_n\}$ and $V = \{v_1, \cdots, v_n\}$, the LLM stores this series of knowledge if the condition $WK = V$ is satisfied. Let knowledge triple $(s_i, r_i, o_i)$ be the $i_{th}$ editing knowledge,

$\{x_j | 0 \le j \le P\}$ be the random token prefixes, $Enc_W(\cdot)$ be the input encoding of the last subject token to $W$. The encoding forms of $k$ and $v$ are as follows.

$$k_i = \frac{1}{P} \sum_{j=1}^{P} Enc_W(x_j \oplus s_i), \tag{1}$$

$$v_i = Wk_i + \delta_i. \tag{2}$$

Here, $\delta_i$ serves as the learnable incremental update applied to the original output of $W$, referred to as the "incremental update parameter". In addition to enable the model to remember the edited knowledge, it is also necessary to preserve the previously stored knowledge pair $(K_0, V_0)$. We randomly sample triples from Wikipedia to encode $K_0$ and obtain $V_0 = WK_0$ [44]. Therefore, after computing $K$ and $V$, an update matrix $\Delta$ is introduced to modify the original parameters $W$ in order to satisfy the desired mapping relationship as follows:

$$(W + \Delta)K = V, (W + \Delta)K_0 = V_0 \tag{3}$$

By applying the normal equations [53, Chapter 4.3], the close-form solution of Eq. 3 is:

$$\Delta = (V - WK)K^T(K_0 K_0^T + KK^T)^{-1}. \tag{4}$$

## 2.2 Overview

Our sequential editing method HSE designs several hippocampal-like mechanisms to precisely modify the model's parameters. As shown in Fig. 1, these mechanisms support memory-directed active forgetting, memory stability preservation to optimize the encoding of editing knowledge, and also progressive memory consolidation to replay existing edited knowledge.

(1) **Memory-directed Active Forgetting** (Sec.2.3) leverages machine unlearning to actively forget old knowledge within the model that is inconsistent with editing knowledge. (2) **Memory Stability Preservation** (Sec.2.4) utilizes the Fisher information matrix to control the updates of model parameters, avoiding mutual interference between editing data from different domains to preserve model stability. (3) **Progressive Memory Consolidation** (Sec.2.5) achieves long-term memory in the model through the progressive replay of edited knowledge using long-term editing memory.

## 2.3 Memory-directed Active Forgetting via Machine Unlearning

To address the conflicting between new editing knowledge and existing knowledge LLMs have learned [34], we propose to leverage machine unlearning to achieve active forgetting of outdated knowledge. The conflict between new and old knowoedge occurs when LLMs edit the data that is inconsistent with the internal knowledge of LLMs. The memory processes in the hippocampus indicate that both memorizing and active forgetting are key mechanisms for the brain to function efficiently [3, 4]. However, current model editing efforts primarily focus on enhancing the memorizing ability [35], with less attention paid to active forgetting. Fortunately, by incorporating machine unlearning techniques, the LLMs can selectively forget outdated knowledge while efficiently acquiring and integrating new knowledge, thereby enhancing its adaptability and performance in dynamic environments.

Specifically, for the editing data $(x, y) \in D$ and corresponding data points $(x, \widetilde{y}) \in \widetilde{D}$ that need to be forgotten, we model the forgetting objective using $1 - p_\delta(\widetilde{y}|x)$ to appropriately reduce the probability of retaining old knowledge within the LLMs. We first construct the maximum likelihood estimation (MLE) optimization to formulate the memorizing and active forgetting objectives for the incremental update parameter $\delta$ in Eq. 2, as shown in Eq. 5, where $\alpha$ balances the degree of memory and forgetting.

$$\delta^* = \arg\min_{\delta} \{ -\alpha \underbrace{\sum_{(x,y) \in D} \log p_\delta(y|x)}_{editing} - (1 - \alpha) \underbrace{\sum_{(x,\widetilde{y}) \in \widetilde{D}} \log[1 - p_\delta(\widetilde{y}|x)]}_{machine\ unlearning} \} \tag{5}$$

Subsequently, we employ the unlikelihood loss [63] to achieve the objective in Eq. 5. We define the loss of $\theta_\delta$ is $\mathcal{L}_{MAF}(\theta_\delta)$. The cross-entropy loss and the unlikelihood loss is denoted as $\mathcal{L}_{CE}(\theta_\delta)$ and $\mathcal{L}_{UL}(\theta_\delta)$ respectively. The $\mathcal{L}_{MAF}(\theta_\delta)$ can be defined as the negative of Eq. 5:

$$\mathcal{L}_{MAF}(\delta) = \mathcal{L}_{CE}(\delta) + \mathcal{L}_{UL}(\delta) = -\frac{\alpha}{|D|} \sum_{(x,y) \in D} \log p_\delta(y|x) - \frac{(1-\alpha)}{|\widetilde{D}|} \sum_{(x,\widetilde{y}) \in \widetilde{D}} \log \left[1 - p_\delta(\widetilde{y}|x)\right] \quad (6)$$

When the optimizer is Adam, we theoretically explain that the unlikelihood loss leads to a reduction in the generalization error bound. This indicates that our method, after editing LLMs, can effectively adapt to more generalized scenarios for editing tasks.

Following the generalization error bounds theorem established in the work [31, Theorem 5.5], we state the following Lemma and Corollary:

**Lemma 1** ([31],Theorem 5.5). *Consider a loss function $\mathcal{L}$ such that $0 \le \mathcal{L}(p, \mathbf{y}) \le L$ and $\gamma$-Lipschitz with respect to the output distribution $p$ and ground-truth label $\mathbf{y}$. Suppose that the Adam optimizer with stabilization constant $c \in (0,1)$ is executed for $T$ iterations with an initial random parameters $\mathcal{R}$, training set $S = \{(x_i, y_i)_{i=1}^N\}$, batch data $B = \{(x_i, y_i)_{i=1}^b\}$ and learning rate $\lambda$ to obtain $f_{B_S,R}$. The empirical risk $R_{emp}$ is defined as $R_{emp}(f_{B_S,\mathcal{R}}) = \frac{1}{b} \sum_{i=1}^b \mathcal{L}(f(x_i), y_i)$ on a finite training set. The true risk $R_{true}(f_{B_S,\mathcal{R}})$ is estimated with the empirical risk over the whole dataset that follows the distribution of the training set. The generalization error $E(f_{B_S,\mathcal{R}}) = R_{true}(f_{B_S,\mathcal{R}}) - R_{emp}(f_{B_S,\mathcal{R}})$. Then, we have the following generalization error bound with probability at least $1 - \epsilon$ :*

$$E(f_{B_S,R}) \le \frac{2\eta}{c} \left(4\left(\frac{b\gamma}{N}\right)^2 \sqrt{T \log(2/\epsilon)} + \frac{bT\gamma^2}{N}(1 + \sqrt{2N \log(2/\epsilon)})\right) + L\sqrt{\frac{\log(2\epsilon)}{2N}}. \quad (7)$$

The proof of Lemma 1 is in [31]. The following corollary compares the generalization error bounds for the learning algorithm which uses CE and MAF loss.

**Corollary 1.** *Consider LLMs are trained using the CE loss and MAF loss separately over the same training set $S$, batch data $B_S$ and other settings. Denote $f_{B_S,\mathcal{R}}^{CE}, f_{B_S,\mathcal{R}}^{MAF}$ as the corresponding LLMs using CE loss and MAF loss. We have the following inequalities:*

$$E\left(f_{B_S,\mathcal{R}}^{MAF}\right) \le E\left(f_{B_S,\mathcal{R}}^{CE}\right) \quad (8)$$

The complete proof of Corollary 1 is provided in Appx. C.1. Corollary 1 provides a tighter bound on the generalization error of Adam when we use MAF loss. Specifically, the generalization error of LLMs trained with MAF loss function is upper-bounded by the generalization error of LLMs trained with CE loss. This result demonstrates that MAF loss can achieve better generalization performance compared to the CE loss. This indicates that our method, after editing for specific queries, can adapt effectively to more generalized scenarios.

## 2.4 Memory Stability Preservation via Fisher Information Matrix

To address the LLMs parameter drift caused by inconsistencies among editing data, we design a method for preserving memory stability using weight matrix (i.e. Fisher information matrix). During the sequential editing process, the cumulative shifts in editing data from different domains also tend to lead parameter drifts in the LLMs. Existing model editing methods primarily focus on the single-edit outcome, and ignore preserving parameter stability [40]. During the formation and stabilization of memorizing in the hippocampus, synaptic consolidation strengthens the connections between specific neurons to protect the stability of important memories [9]. Therefore, we employ the calculated weighted Fisher information matrix (FIM) [29, 23] to protect the LLMs. FIM constrains the update magnitude of parameters with significant influence on the model outputs, and permits more substantial updates for parameters with less impact on the model outputs.

Specifically, we update the parameter $\delta$ in Eq. 2 to avoid affecting the existing knowledge, instead of directly constraining the updates of the parameter $W$. The motivation is we observed that, in sequential editing tasks, due to the domain gaps between the editing data, training $\delta$ tends to lead parameter drift and model instability. For the sequential editing data $e_1 = (s_1, r_1, o_1, o_1^*)$ and $e_2 = (s_2, r_2, o_2, o_2^*)$, we need to satisfy both $e_1$ and $e_2$ simultaneously. Thus, $\delta_{e_1,e_2}$ is obtained by using the Maximum A Posteriori (MAP) estimate and Bayes' theorem:

$$\delta_{e_1,e_2}^* = \arg\max_\delta p(\delta|e_1, e_2) = \arg\max_\delta \frac{p(e_2|\delta)p(\delta|e_1)}{p(e_2)}. \quad (9)$$

Since we optimize $\delta$, the probability $p(e_2)$ remains constant, indicating the distribution of the editing data $e_2$ independent of $\delta$. Therefore, the objective of $\delta^*_{e_1,e_2}$ is :

$$\delta^*_{e_1,e_2} = \arg\max_\delta p(e_2|\delta)p(\delta|e_1) = \arg\max_\delta[\log p(e_2|\delta) + \log p(\delta|e_1)] \tag{10}$$

The term $\log p(e_2|\delta)$ corresponds to the objective of the negative loss function $-\mathcal{L}_{e_2}(\delta)$ in Eq. 6. Thus, the optimization objective is further refined as follows:

$$\delta^*_{e_1,e_2} = \arg\max_\delta[-\mathcal{L}_{e_2}(\delta) + \log p(\delta|e_1)] = \arg\min_\delta[\mathcal{L}_{e_2}(\delta) - \log p(\delta|e_1)]. \tag{11}$$

However, the term $p(\delta|e_1)$ lacks a straightforward or interpretable form, which is intractable to compute its quantiles [29]. Laplace approximation [41] approximates complex posterior probability distributions using a Gaussian distribution. This approximation assumes that the posterior probability distribution can be approximated by a quadratic function around its maximum. Therefore, we utilize this approach to approximate the probability $p(\delta|e_i)$ with a Gaussian distribution in general cases, leading to the following Theorem 1.

**Theorem 1.** *Assume that the $i_{th}$ editing data $e_i = (s_i, r_i, o_i, o_i^*)$ and the posterior probability $p(\delta|e_i)$ is smooth which reaches its maximum around $\delta^*_{e_i}$. Then, it can be approximated by a Gaussian distribution $\mathcal{N}$ with mean $\delta^*_{e_i}$ and variance $F_{e_i}^{-1}$, where $F_{e_i}$ is the Fisher information matrix of the LLM after editing $e_i$. Specifically, the approximation is given by:*

$$p(\delta|e_i) \sim \mathcal{N}\left(\delta^*_{e_i}, F_{e_i}^{-1}\right) \tag{12}$$

$$F_{e_i} = \mathbb{E}\left[\left(\frac{\partial \log p(\delta|e_i)}{\partial \delta}\right)\left(\frac{\partial \log p(\delta|e_i)}{\partial \delta}\right)^\top \bigg|_{\delta^*_{e_i}}\right]. \tag{13}$$

The detail of proof is in Appx. C.2. According to work [1], we obtain $F_{e_i}$ by treating $p(e_i)$ constant and ignoring the prior $p(\delta)$, which makes derivative of posterior $\log p(\delta|e_i)$ and likelihood $\log p(e_i|\delta)$ equal. Based on the Gaussian distribution 12 and Eq. 2 assuming that $\delta^*_{e_i} = \mathbf{0}$ for the $(i+1)_{th}$ edit, we obtain $\log p(\delta|e_i)$,

$$\log p(\delta|e_i) = \frac{\lambda_i}{2}\delta^T F_{e_i}\delta. \tag{14}$$

For scenarios involving more than two editing data, we obtain the posterior probability:

$$\log(\delta|e_1, e_2, \cdots, e_{n-1}) = \frac{1}{2}\delta^T \sum_{i=1}^{n-1}(\lambda_i F_{e_i})\delta. \tag{15}$$

Therefore, based on Eq. 6, 11 and 15, for $n$ times editing we formulate total training loss for $\delta$ as:

$$\mathcal{L}_\delta = \mathcal{L}_{MAF} + \frac{1}{2}\delta^T \sum_{i=1}^{n-1}(\lambda_i F_{e_i})\delta, \tag{16}$$

where $\lambda_i$ are hyperparameters that control the magnitude of the Fisher information matrix loss, $n$ is the number of sequential editing. We leverage $F_{e_i}$ to precisely control the magnitude of changes in each parameter of $\delta$, which differs from conventional weight decay methods. Eq. 16 shows that parameters with higher Fisher information values exhibit lower update magnitudes when exposed to new editing data. This prevents interference between data from different domains and facilitates an effective separation mechanism. Note that Sec. 2.4 further accounts for interference among different edits, which is not in conflict with the modeling approach presented in Sec. 2.3, but rather constitutes a complementary extension.

## 2.5 Progressive Memory Consolidation via Parameter Replay

Sequential editing of LLMs is prone to catastrophic forgetting of existing knowledge. Thus, we utilize long-term editing memory to progressively replay edited knowledge. When we edit LLMs to learn new knowledge, the lack of consolidation of old knowledge disrupts previously learned representations. Memory consolidation in the hippocampus, which transforms short-term memories into long-term memories, relies on the reactivation and enhancement of neural activity patterns associated with previously experienced events through CA3→CA1 sharp wave ripples [26, 13]. This process requires repeated reinforcement until the corresponding memory is stored in the cerebral cortex [28]. In

typical model editing method, only short-term memory retains recently edited knowledge. Inspired by hippocampus mechanism, we derive a rigorous closed-form solution to maintain a "long-term editing memory" that replays all edited knowledge.

Specifically, according to Eq. 4, we note that the incremental matrix $\Delta$ considers only the knowledge pair $(K, V)$ in single edit operation. In the sequential editing scenario, we extend this to $n$ edits to obtain the following theorem:

**Theorem 2.** *Assume that $W$ after the $i_{th}$ sequential edit is denoted as $W_i$. The knowledge pairs associated with the $i_{th}$ edit is represented by keys $K_i$ and values $V_i$. Let $C_0 = \lambda_C K_0 K_0^T$, $\delta_i = V_i - W_{i-1} K_i$ and $Cov\_sum_{i-1} = \sum_{j=1}^{i-1} K_j K_j^T$. $\lambda_C$ is hyperparameter. The convergence factor $\alpha_i = \frac{n}{i-1}$ $(i > 1)$ ensures the convergence of the sum of $\Delta_i$ and balances the degree of consolidation for different editing knowledge. Then it follows that:*

$$W_i = W_{i-1} + \Delta_i \tag{17}$$

$$\Delta_i = \delta_i K_i^T (C_0 + \alpha_i Cov\_sum_{i-1} + K_i K_i^T)^{-1}, \tag{18}$$

**Corollary 2** (Convergence). *Given $\Delta_i$ as defined in Theorem 2, let $\alpha_i = \frac{n}{i-1}$ $(i > 1)$ and the minimum eigenvalue of $C_0$ and the maximum eigenvalue of $K_i K_i^T$ $(K_i \in \mathbb{R}^q)$ are all at least 1. Assume that for all indices $i \leq q$, $K_i$ are mutually orthogonal (practical). Then the Frobenius norm $\lim_{n \to \infty} ||W_n||_F$ converges.*

The derivation of long-term editing memory in Theorem 2 and the proof of the convergence factor $\alpha_i$ in Corollary 2 are in Appx. C.3. In this theorem, $Cov\_sum$ represents the long-term editing memory, which is the sum of the covariances of all non-normalized keys $K$. Since the $Cov\_sum = \sum KK^T$ encapsulate all the input information from previous edits, we consider this is a form of parameter replay. Corollary 2 demonstrates that the introduced convergence factor effectively prevents the Frobenius norm of the edited parameters from surge. This approach does not require substantial storage or computational resources, which is in contrast to traditional replay mechanisms [50]. By employing an iterative method, we only need to compute the $Cov\_sum$ by accumulating the $KK^T$ from the previous editing step. The overall procedure of the HSE method is detailed in the Appx. B

## 3 Experiments

### 3.1 Performance Results on Sequential Editing

In this section, we compare our proposed method HSE, against other leading model editing baselines on two benchmark datasets. For detailed descriptions of the datasets, baselines, implementation and evaluation metrics, please see Appx. E.1, E.2 and E.3.

To validate the generalization of our editing approach, we conduct one-by-one sequential editing with 1,000 samples experiments on four open-source LLMs respectively: Llama3-Instruct (8B), Mistral-7B-Instruct-V0.3, GPT-J (6B), and GPT2-XL (1.5B). As shown in Fig. 2 (a)-(d), the HSE method achieves superior performance across all LLMs and datasets. Compared to the best baseline, our approach demonstrates average improvements of 20.6% in generalization, 21.9% in specificity and 17.3% in efficacy. For all models, most baselines suffer from catastrophic performance degradation due to model parameter collapse after multiple edits. Notably, the Llama3 and Mistral models exhibit particularly significant improvements across various metrics. This is attributed to their smaller F-norm of parameters, which makes other methods more susceptible to catastrophic forgetting (Fig. 9). Additionally, HSE exhibits the highest fluency and consistency, even surpassing the original model in multiple consistency evaluations. The enhanced specificity performance may be attributed to the fact that during the editing process, the LLMs revisit and refine their knowledge about the subject, leading to a deeper understanding. The excellent performance on both counterfactual and standard QA datasets demonstrates the effectiveness of our proposed approach.

We do not include a comparison with the AlphaEdit [10] and F-learning [47] baselines in the figure. Instead, a more detailed comparison and discussion are provided in Appx. D and E.7. In addition, we provide case study in Appx. F and time complexity analysis and performance comparison across various batch sizes for the full-batch editing scenario in the Appx. E.6.

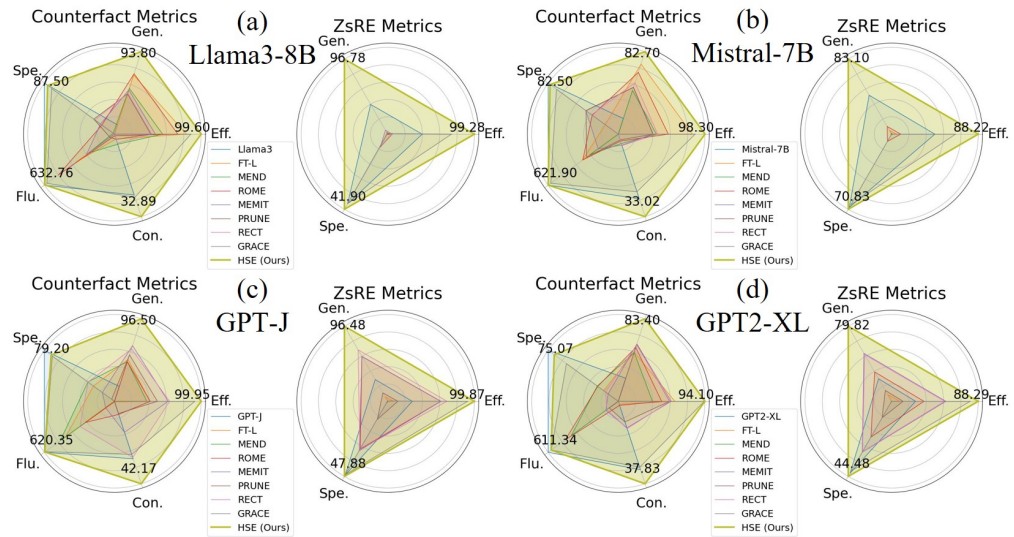

Figure 2: **(a)**-**(d)** present the performance comparison results of one-by-one sequential editing across four LLMs and two datasets Counterfact and ZsRE , using 1,000 samples. Eff.,Gen.,Spe., Flu. and Con. represent efficacy, generalization, specificity, fluency, and consistency, respectively.

## 3.2 Downstream Evaluation after Sequential Editing

To evaluate the general capabilities of edited LLMs, we tested Llama3 on six tasks from the GLUE benchmark [58] after sequential editing with CounterFact and ZsRE respectively. Details of the GLUE tasks can be found in Sec. 2.

As shown in Fig. 3, our proposed HSE method not only maintains but slightly surpasses the original performance of Llama3 across all six tasks even after 1,000 sequential editing steps. In contrast, other methods exhibit a sharp decline in performance to near zero. As a special instance of HSE, AlphaEdit effectively preserves most of the model's general capabilities, yet still demonstrates a marked decrease in performance. After editing with CounterFact data, the performance of MEND and FT-L methods across all tasks drops to zero within 100 steps. Moreover, MEMIT and PRUNE maintain performance until approximately 400~500 steps before experiencing a sudden decline to zero. Following edits with ZsRE data as shown in Fig. 3, MEND and FT-L methods again exhibit a rapid drop to zero within 100 steps, while MEMIT and PRUNE show a delayed performance collapse, occurring around 600 steps. The reasons for the sudden performance drops of these methods are discussed in the ablation study part (Appx. E.5). Our proposed HSE method exhibits the following behavior. Given that the facts in CounterFact are counter to reality, the edited model may experience a slight 1.42% decrease in general capability during testing. However, when the model is correctly edited with knowledge from the ZsRE QA dataset, its general capability shows an average 1.67% improvement. These results indicate that our editing method HSE minimally impacts the model's general capability and can lead to slight enhancements when guiding the model with correctly edited knowledge.

## 3.3 Sequential Editing Results on Practical Applications

To demonstrate the broad applicability of the proposed HSE method, we conduct experiments on three practical applications: hallucination mitigation (Fig. 4), healthcare knowledge injection (Fig. 5), and societal bias reduction (Fig. 6). Experiments demonstrate that HSE achieves consistently superior performance across the practical applications compared to existing sequential editing methods, highlighting its strong potential for future development and real-world deployment (see Appx. E.4).

## 3.4 Ablation Study to Validate the Hippocampal-like Design

To systematically evaluate the contribution of each component in our framework, we first conduct ablation experiments with intuitive performance comparisons (Tab. 2). Additionally, to justify the

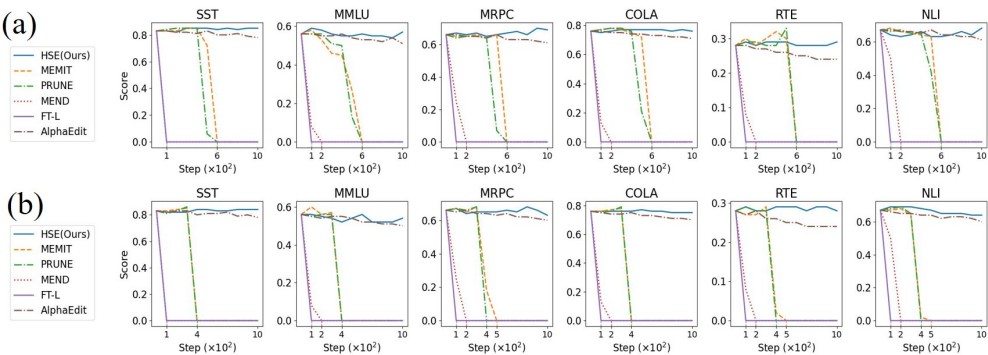

Figure 3: The general capabilities of the Llama3 model on the six tasks of the GLUE benchmark after editing with the CounterFact **(a)** and ZsRE **(b)** datasets respectively.

hippocampal-like design of each module, we carry out further analyses, including visualizations of machine unlearning (Fig. 7), domain-specific knowledge update separation (Fig. 8), and the stability of long-term editing memory during replay (Fig. 9). As shown in the experimental analysis presented in Appx. E.5, our proposed module not only significantly improves performance on sequential editing, but also aligns well with the biological principles underlying memory consolidation in the hippocampus, thereby reinforcing the neuro-inspired foundation of our approach.

Notably, the Long-term Editing Memory (LEM) module plays a pivotal role in mitigating model collapse and catastrophic forgetting. This is evidenced in Fig. 9, where LEM effectively constrains the deviation of the parameter matrix's Frobenius norm, establishing it as a fundamental component compared to other modules. The Fisher Information Matrix (FIM) design serves to quantify the retention degree of prior knowledge. Fig. 8 demonstrates that the FIM successfully safeguards critical parameter updates for edited knowledge while preventing interference across distinct knowledge updates. The Active Forgetting module dynamically regulates knowledge retention versus discarding. Ablation results confirm that AF contributes most significantly to model generalization, a finding that is consistent with the theoretical implications of Corollary 1.

## 4    Related Works

**Sequential Model Editing.** Current approaches to sequential model editing can be broadly categorized into two paradigms: parameter-preserving and parameter-modifying methods. *Parameter-preserving methods* typically introduce auxiliary modules to store edited knowledge [22, 20, 46, 61, 68, 72]. SERAC [46] maintains an explicit memory of edits and employs a classifier to determine whether to apply the stored knowledge during inference. T-patcher [22] proposes sequential model editing by continuously adding and fine-tuning a small set of neurons in the final FFN layer to accommodate new edits. GRACE [20] records edit knowledge as key-value pairs within a learnable codebook, enabling persistent and scalable knowledge updates. WISE [61] introduces a side-memory module that is trained to retain previously edited knowledge. In contrast, *parameter-modifying methods* follow a meta-learning [45, 8] or locate-then-edit paradigm [43, 44, 14, 10, 40], for sequential editing. MEMIT [44] enables efficient batch editing of factual knowledge but tends to suffer from model collapse under sequential editing. Both RECT [14] and PRUNE [40] refine the update matrix through post-processing: RECT regularizes the update, while PRUNE controls its singular values to sequential editing. AlphaEdit [10] orthogonalizes newly injected knowledge with existing knowledge to mitigate catastrophic forgetting. This method is discussed in our work as a special case of our proposed replay mechanism. Most existing approaches to sequential model editing do not adequately mitigate parameter drift that arises when newly edited knowledge conflicts with existing knowledge, often leading to degradation in general capabilities.

**Bio-inspired Continual Learning.** Recent advances in continual learning have increasingly drawn inspiration from biological systems to address catastrophic forgetting [1, 39, 24, 56, 27, 37, 16, 59, 71, 70]. *Neuro-inspired adaptability models*, fruit fly learning mechanisms employ selective memory protection and forgetting to balance stability and plasticity [59]. CLASSP [39] leverage synaptic

plasticity principles by suppressing non-critical weight updates while promoting sparse learning. *Replay-based approaches*, BiRT [24], Replay-through-Feedback [56] and Robust Rehearsal [27], mimic memory reconsolidation in biological systems to refine past experiences. Additionally, GEM [37] and EWC [1] incorporate episodic memory and synaptic consolidation respectively, to mitigate interference between tasks. These biologically inspired strategies highlight the potential of neuroscience principles in advancing continual learning paradigms.

## 5    Conclusion

In this paper, we propose a Hippocampal-like Sequential Editing (HSE) method, which designs the unlearning of obsolete knowledge, domain-specific knowledge update separation and replay for edited knowledge. Our experimental results show that HSE significantly outperforms existing model editing methods across multiple benchmarks. Under sequential editing conditions, the LLMs can almost entirely retain and even enhance its general capabilities. Theoretical analysis confirms that the HSE method exhibits tighter generalization error bounds, more stable convergence and higher computational efficiency. Furthermore, HSE facilitates significant practical advancements in mitigating multi domain hallucinations, updating healthcare knowledge and reducing societal biases for LLMs.

## 6    Acknowledgments

This work was financially supported by the National Natural Science Foundation of China (No. 62476283, No. 62376284).

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

# Technical Appendices and Supplementary Material

## A  Key Parameters and Descriptions

To facilitate the readability and comprehension of our work, we provide a detailed description of the commonly used and key parameters. The specific descriptions are as follows:



### Nomenclature

| | |
|---|---|
| $s$ | Subject of the edited knowledge |
| $r$ | Relation of the edited knowledge |
| $o$ | Object of the original knowledge |
| $o^*$ | Target object of the edited knowledge |
| $x$ | Random token prefixes |
| $k$ | Input encoding of the knowledge before entering edited parameter |
| $v$ | Output encoding of the knowledge after entering edited parameter |
| $\delta$ | Incremental update parameter for $v$ (Eq. 2) |
| $\delta^*_{e_1,e_2}$ | Optimal $\delta$ when editing $e_2$ while jointly considering both $e_1$ and $e_2$ |
| $\Delta$ | Incremental update matrix for modifying the edited parameters (from Theorem 2) |
| $F_e$ | Fisher information matrix of the LLM after editing e |
| $\gamma$ | Lipschitz constant |
| $Cov\_sum$ | Long-term editing memory |
| $\alpha$ | Balance the degree of memory and forgetting |
| $\lambda_i$ | Control the magnitude of the Fisher loss |
| $\lambda_C$ | Control the retention of original knowledge |
| $\alpha_i$ | Convergence factor for preserving general capabilities |



## B  HSE Procedure

To facilitate the practical application of HSE, we present a detailed exposition of its specific algorithms and operational procedures. We illustrate the editing process using the most common scenario, one-by-one sequential editing. The process is illustrated in the algorithm 1:

For the localization operation in the algorithm, we employ established methods [44, 43] to perform causal tracing for noise recovery and identify the edited parameters. These parameters are typically located in the second linear layer of the FFN within the early transformer modules [57] of LLMs. In this procedure, we do not explicitly illustrate the editing process for different layers. The residual spread between layers is implemented using the formula $\frac{\delta_L}{L-l-1}$, where $L$ denotes the last edited layer and $l$ represents other edited layers. Detailed residual spread details can be found in the provided code and work [44].

## C  Proof of Main Results

### C.1  Proof of Corollary 1

**Notation.** Let $p \in \mathbb{R}^{|\mathcal{V}|}$ be the output probability distribution of edited LLM. Let $p_{pos}$ and $p_{neg}$ be the positive and negative next-token probability (index $i_{pos}, i_{neg} \in \mathcal{V}$) respectively. Let $\mathbf{y}$ denote ground-truth label. $\|\cdot\|_2$ is denoted as the $\ell_2$-normalization. Considering an example $(x, y)$ to be edited and the corresponding sample $(x, \widetilde{y})$ to be forgotten, , and for simplicity ignoring the normalization factors $|D|$ and $|\widetilde{D}|$, we derive the gradients of $\mathcal{L}_{CE}$ and $\mathcal{L}_{MAF}$ with respect to $p$:

$$\|\nabla \mathcal{L}_{CE}(p, \mathbf{y})\|_2^2 = \left\|\frac{\partial \mathcal{L}_{CE}}{\partial p}\right\|_2^2 = \frac{1}{p_{pos}^2}, \tag{19}$$

---

**Algorithm 1** HSE algorithm

---

**Require:** Initial LLM $f_0$, **Editing dataset** $\mathcal{D} = \{e_i = (s_i, r_i, \widetilde{o}_i, o_i^*)\}_{i=1}^n$, **Templated prompt** $p(\cdot)$,
    **Random token prefixes ditribution** $\mathcal{U}_{\text{prefix}}$,**Wikitext** $\mathcal{T}$, **Hyperparameters** $\lambda_C, \{\alpha_i\}_{i=1}^n$

**Ensure:** $f_n$

  1: **Locate edited parameters in LLM:**
  2:     $W_0 \leftarrow Causal\ trace(f_0)$
  3: **Get general knowledge in original LLM:**
  4:     $K_0 \leftarrow \text{SampleWiki}(\mathcal{T}, 10^5)$
  5:     $C_0 \leftarrow \lambda_C K_0 K_0^T$
  6: **Initialize long-term editing memory:**
  7:     $Cov\_sum_0 \leftarrow 0$
  8:     $F_{e_1} \leftarrow 0$
  9: **for** $i = 1$ **to** $n$ **do**
10:     $\{x_j\}_{j=1}^P \overset{\text{iid}}{\sim} \mathcal{U}_{\text{prefix}}$
11:     **Compute key vector:**
12:         $k_i \leftarrow \frac{1}{P} \sum_{j=1}^P Enc_{W_i}(x_j \oplus s_i)$
13:     **Compute value vector:**
14:         $v_i \leftarrow W_{i-1} k_i + \delta_i$
15:         $\delta_i \leftarrow \arg\min_\delta -\alpha \log p_\delta(o_i^* | p(s_i, r_i)) - (1-\alpha) \log\left[1 - p_\delta(\widetilde{o}_i | p(s_i, r_i))\right] + \frac{1}{2}\delta^T \sum_{j=1}^{i-1} \left(\lambda_j F_{e_j}\right) \delta$
16:         $F_{e_i} \leftarrow \mathbb{E}\left[ \left(\frac{\partial \log p(\delta | e_i)}{\partial \delta}\right) \left(\frac{\partial \log p(\delta | e_i)}{\partial \delta}\right)^\top \bigg|_{\delta_i} \right]$
17:     **Compute update:**
18:         $\Delta_i \leftarrow \delta_i k_i^\top (C_0 + \alpha_i Cov\_sum_{i-1} + k_i k_i^\top)^{-1}$
19:     **Long-term editing memory update:**
20:         $Cov\_sum_i \leftarrow Cov\_sum_{i-1} + k_i k_i^T$
21:     **Apply edit:**
22:         $W_i \leftarrow W_{i-1} + \Delta_i$
23: **end for**
24: $f_n \leftarrow Replace(W_n, f_0)$

---

$$\|\nabla \mathcal{L}_{MAF}(p, \mathbf{y})\|_2^2 = \left\|\frac{\partial \mathcal{L}_{MAF}}{\partial p}\right\|_2^2 = \frac{\alpha^2}{p_{pos}^2} + \frac{(1-\alpha)^2}{(1-p_{neg})^2}. \tag{20}$$

**Proof Overview.** According to Definition 1, we establish that both $\mathcal{L}_{CE}$ and $\mathcal{L}_{MAF}$ satisfy the Lipschitz property during training. By Corollary 2, we find that the Lipschitz constant $\gamma_{\text{MAF}}$ for $\mathcal{L}_{MAF}$ is smaller than the Lipschitz constant $\gamma_{\text{CE}}$ for $\mathcal{L}_{CE}$. According to Lemma 1 , we know that the generalization error bound is influenced by $\gamma$ and the maximum value of the loss $L$. In the subsequent Corollary 1,we demonstrate that the proposed loss $\mathcal{L}_{MAF}$ has a tighter generalization error bound compared to $\mathcal{L}_{CE}$.

**Definition 1** (Lipschitz Property). *A loss function $\mathcal{L}(p, \mathbf{y})$ is $\gamma$-Lipschitz (admissible) with respect to the p, if for $\gamma \geq 0$ and $\forall p_1, p_2 \in \mathbb{R}^{|\mathcal{V}|}$ we have*

$$|\mathcal{L}(p_1, \mathbf{y}) - \mathcal{L}(p_2, \mathbf{y})| \leq \gamma \|p_1 - p_2\|_2. \tag{21}$$

From Eq. 19, 20 and a practical point of view, when training such that $p_{pos} > 0.5$ and $p_{neg} < 0.5$, we observe that both $\|\nabla \mathcal{L}_{CE}(p, \mathbf{y})\|_2^2$ and $\|\nabla \mathcal{L}_{MAF}(p, \mathbf{y})\|_2^2$ are less than 4. This implies that under this condition, both $\mathcal{L}_{CE}$ and $\mathcal{L}_{MAF}$ satisfy Lipschitz Property. Therefore, all subsequent discussions are based on this condition.

**Lemma 2.** *Given that the CE and MAF loss functions are $\gamma_{\text{CE}}$-Lipschitz, $\gamma_{\text{MAF}}$-Lipschitz with respect to the output distribution p, with respective upper bounds $L_{CE}$ and $L_{MAF}$, the following inequality holds:*

$$\gamma_{\text{MAF}} \leq \gamma_{\text{CE}}, \tag{22}$$

$$L_{MAF} \leq L_{CE}. \tag{23}$$

*Proof.* According to the Eq. 19 and 20, we subtract the two equations to get

$$\Delta^p = \|\nabla \mathcal{L}_{MAF}(p, \mathbf{y})\|_2^2 - \|\nabla \mathcal{L}_{CE}(p, \mathbf{y})\|_2^2 = \frac{(\alpha^2 - 1)}{p_{pos}^2} + \frac{(1 - \alpha)^2}{(1 - p_{neg})^2}. \tag{24}$$

From the properties of the probability distribution, we have $p_{pos} + p_{neg} \leq 1$. Therefore, it follows that $\frac{1}{1 - p_{neg}} \leq \frac{1}{p_{pos}}$. Consequently, we can derive:

$$\begin{aligned}
\Delta^p &= \frac{(\alpha^2 - 1)}{p_{pos}^2} + \frac{(1 - \alpha)^2}{(1 - p_{neg})^2} \\
&\leq \frac{(\alpha^2 - 1)}{p_{pos}^2} + \frac{(1 - \alpha)^2}{p_{pos}^2} \\
&= \frac{\alpha^2 - 1 + (1 - \alpha)^2}{p_{pos}^2} \\
&= \frac{2\alpha(\alpha - 1)}{p_{pos}^2} \leq 0.
\end{aligned} \tag{25} \tag{26}$$

From the sign of $\Delta^p$, we can conclude that $\|\nabla \mathcal{L}_{MAF}(p, \mathbf{y})\|_2^2$ is less than $\|\nabla \mathcal{L}_{CE}(p, \mathbf{y})\|_2^2$ which means $\gamma_{\mathrm{MAF}} \leq \gamma_{\mathrm{CE}}$. Because $p_{pos} + p_{neg} < 1$, it follows that $-\log(1 - p_{neg}) < -\log p_{pos}$, we derive that

$$\begin{aligned}
\mathcal{L}_{MAF} &= -\alpha \log p_{pos} - (1 - \alpha) \log (1 - p_{neg}) \\
&\leq -\alpha \log p_{pos} - (1 - \alpha) \log p_{pos} \\
&= -\log p_{pos} = \mathcal{L}_{CE}.
\end{aligned} \tag{27}$$

Therefore, from relation 26 and 27, we prove that $\gamma_{\mathrm{MAF}} \leq \gamma_{\mathrm{CE}}$ and $\mathcal{L}_{MAF} \leq \mathcal{L}_{CE}$. □

**Lemma 1** ([31],Theorem 5.5). *Consider a loss function $\mathcal{L}$ such that $0 \leq \mathcal{L}(p, \mathbf{y}) \leq L$ and $\gamma$-Lipschitz with respect to the output distribution $p$ and ground-truth label $\mathbf{y}$. Suppose that the Adam optimizer with stabilization constant $c \in (0, 1)$ is executed for $T$ iterations with an initial random parameters $\mathcal{R}$, training set $S = \{(x_i, y_i)_{i=1}^N\}$, batch data $B = \{(x_i, y_i)_{i=1}^b\}$ and learning rate $\lambda$ to obtain $f_{B_S, R}$. The empirical risk $R_{emp}$ is defined as $R_{emp}(f_{B_S, \mathcal{R}}) = \frac{1}{b} \sum_{i=1}^b \mathcal{L}(f(x_i), y_i)$ on a finite training set. The true risk $R_{true}(f_{B_S, \mathcal{R}})$ is estimated with the empirical risk over the whole dataset that follows the distribution of the training set. The generalization error $E(f_{B_S, \mathcal{R}}) = R_{true}(f_{B_S, \mathcal{R}}) - R_{emp}(f_{B_S, \mathcal{R}})$. Then, we have the following generalization error bound with probability at least $1 - \epsilon$ :*

$$E(f_{B_S, R}) \leq \frac{2\eta}{c} \left( 4 \left( \frac{b\gamma}{N} \right)^2 \sqrt{T \log(2/\epsilon)} + \frac{bT\gamma^2}{N} (1 + \sqrt{2N \log(2/\epsilon)}) \right) + L\sqrt{\frac{\log(2\epsilon)}{2N}}. \tag{28}$$

**Corollary 1.** *Consider LLMs are trained using the CE loss and MAF loss separately over the same training set $S$, batch data $B_S$ and other settings. Denote $f_{B_S, \mathcal{R}}^{\mathrm{CE}}, f_{B_S, \mathcal{R}}^{\mathrm{MAF}}$ as the corresponding LLMs using CE loss and MAF loss. We have the following inequalities:*

$$E\left(f_{B_S, \mathcal{R}}^{\mathrm{MAF}}\right) \leq E\left(f_{B_S, \mathcal{R}}^{\mathrm{CE}}\right) \tag{29}$$

*Proof.* As shown in Lemma 1, the generalization error bounds for different loss functions under the same training process are determined by $\gamma$ and $L$. Given that $\gamma_{MAF} \leq \gamma_{CE}$ in inequality 22, we know that the first term in the generalization error bound for MAF (as given in inequality 7) is smaller. Additionally, inequality 23 indicates that the second term in inequality 7 of MAF loss is smaller than the corresponding term of CE loss. Hence, we can conclude that $E\left(f_{B_S, \mathcal{R}}^{\mathrm{MAF}}\right) \leq E\left(f_{B_S, \mathcal{R}}^{\mathrm{CE}}\right)$. □

### C.2 Derivation of Fisher Information Matrix for $\delta$

**Theorem 1.** *Assume that the $i_{th}$ editing data $e_i = (s_i, r_i, o_i, o_i^*)$ and the posterior probability $p(\delta|e_i)$ is smooth which reaches its maximum around $\delta_{e_i}^*$. Then, it can be approximated by a Gaussian distribution $\mathcal{N}$ with mean $\delta_{e_i}^*$ and variance $F_{e_i}^{-1}$, where $F_{e_i}$ is the Fisher information matrix of the LLM after editing $e_i$. Specifically, the approximation is given by:*

$$p(\delta|e_i) \sim \mathcal{N}\left(\delta_{e_i}^*, F_{e_i}^{-1}\right) \tag{30}$$

$$F_{e_i} = \mathbb{E}\left[\left(\frac{\partial \log p(\delta|e_i)}{\partial \delta}\right)\left(\frac{\partial \log p(\delta|e_i)}{\partial \delta}\right)^T \bigg|_{\delta_{e_i}^*}\right]. \tag{31}$$

*Proof.* Given the editing data $e_i$ and $e_{i+1}$, according to Eq. 11, we have

$$\delta^*_{e_i, e_{i+1}} = \arg \min_\delta [\mathcal{L}_{e_{i+1}}(\delta) - \log p(\delta|e_i))]. \tag{32}$$

We define $\delta^*_{e_i} = \arg \max_\delta p(\delta|e_i)$ peaked around its point of maxima. Let $f(\delta) = \log p(\delta|e_i)$ and the first derivative satisfies $\frac{\partial f(\delta)}{\partial \delta}|_{\delta^*_{e_i}} = 0$. Therefore, the second-order Taylor expansion of $f(\delta)$ around $\delta^*_{e_i}$ is given by

$$f(\delta) \approx f(\delta^*_{e_i}) + \frac{1}{2}(\delta - \delta^*_{e_i})^\top \underbrace{\left( \frac{\partial^2 f(\delta)}{\partial^2 \delta} \bigg|_{\delta^*_{e_i}} \right)}_{\text{Hessian}} (\delta - \delta^*_{e_i}). \tag{33}$$

Next, since $f(\delta^*_{e_i})$ can be treated as a constant, we express the Hessian matrix as $-\left( \left( -\frac{\partial^2 f(\delta)}{\partial^2 \delta} \bigg|_{\delta^*_{e_i}} \right)^{-1} \right)^{-1}$. Based on the properties of the Fisher information matrix [48], we know that

$$E\left[ -\frac{\partial^2 f(\delta)}{\partial^2 \delta} \bigg|_{\delta^*_{e_i}} \right] = E\left[ \left( \left( \frac{\partial f(\delta)}{\partial \delta} \right) \left( \frac{\partial f(\delta)}{\partial \delta} \right)^\top \right) \bigg|_{\delta^*_{e_i}} \right] = F_{e_i}. \tag{34}$$

Therefore, from Eq. 33 and 34, we obtain

$$p(\delta|e_i) = \exp f(\delta^*_{e_i}) \cdot \exp\left( -\frac{1}{2} \left( \delta - \delta^*_{e_i} \right)^\top F_{e_i} \left( \delta - \delta^*_{e_i} \right) \right). \tag{35}$$

Since Eq. 35 conforms to the form of a Gaussian distribution, we derive the Laplace approximation of the posterior probability

$$p(\delta|e_i) \sim \mathcal{N}\left( \delta^*_{e_i}, F_{e_i}^{-1} \right). \tag{36}$$

From results 34 and 36, we can conclude that Theorem 1 holds. $\square$

## C.3 Derivation of $\Delta$ and Long-term Editing Memory

The derivation and proof of and long-term editing memory, as presented in Theorem 2, are as follows.

**Theorem 2.** *Assume that $W$ after the $i_{th}$ sequential edit is denoted as $W_i$. The knowledge pairs associated with the $i_{th}$ edit is represented by keys $K_i$ and values $V_i$. Let $C_0 = \lambda_C K_0 K_0^T$, $\delta_i = V_i - W_{i-1} K_i$ and $Cov\_sum_{i-1} = \sum_{j=1}^{i-1} K_j K_j^T$. $\lambda_C$ is hyperparameter. The convergence factor $\alpha_i = \frac{n}{i-1}$ $(i > 1)$ ensures the convergence of the sum of $\Delta_i$ and balances the degree of consolidation for different editing knowledge. Then it follows that:*

$$W_i = W_{i-1} + \Delta_i \tag{37}$$

$$\Delta_i = \delta_i K_i^T (C_0 + \alpha_i Cov\_sum_{i-1} + K_i K_i^T)^{-1}, \tag{38}$$

*Proof.* **Step 1 Base Case.** When $i = 1$, base case in Eq. 4 has been proved, so we only need to prove the Inductive Step.

**Step 2 Induction Hypothesis.** Assume that $i = n - 1$, the optimal parameter $W_{n-1}$ through $n - 1$ times sequential editing satisfies

$$W_{n-1} K_0 = V_0$$
$$W_{n-1} K_1 = V_1$$
$$\dots$$
$$W_{n-1} K_{n-1} = V_{n-1} \tag{39}$$

**Step 3 Inductive Step.** When $i = n$, the ideal optimal parameter $W_n$ is supposed to satisfy one additional condition $W_n K_n = V_n$. Then

$$W_n \begin{bmatrix} K_0 \dots K_n \end{bmatrix} = \begin{bmatrix} V_0 \dots V_n \end{bmatrix} \tag{40}$$

$$\text{normal equation:} \quad W_n \begin{bmatrix} K_0 \dots K_n \end{bmatrix} \begin{bmatrix} K_0 \dots K_n \end{bmatrix}^T = \begin{bmatrix} V_0 \dots V_n \end{bmatrix} \begin{bmatrix} K_0 \dots K_n \end{bmatrix}^T \tag{41}$$

$$\text{which expands to:} \quad (W_{n-1} + \Delta_n)(K_0 K_0^T + \dots + K_n K_n^T) = V_0 K_0^T + \dots + V_n K_n^T \tag{42}$$

$$W_{n-1}(K_0 K_0^T + \dots + K_n K_n^T) + \Delta_n(K_0 K_0^T + \dots + K_n K_n^T) = V_0 K_0^T + \dots + V_n K_n^T \tag{43}$$

from induction hypothesis 39 we know that $W_{n-1}K_iK_i^T = V_iK_i^T (0 \le i \le n-1)$, so we can simplify the Equation 43 as

$$W_{n-1}K_nK_n^T + \Delta_n(K_0K_0^T + \cdots + K_nK_n^T) = V_nK_n^T \tag{44}$$

$$\Delta_n(K_0K_0^T + \cdots + K_nK_n^T) = V_nK_n^T - W_{n-1}K_nK_n^T \tag{45}$$

$$\Delta_n = \delta_nK_n^T(K_0K_0^T + \cdots + K_iK_i^T + \cdots + K_nK_n^T)^{-1}. \tag{46}$$

Given that each equation in inductive hypothesis 39 can have distinct factor, the term $K_iK_i^T$ in Eq. 46 share the same factor. For briefness, we set the convergence factor $\alpha_i$ and variable $Cov\_sum_i = \sum_{j=1}^{i-1} K_iK_i^T$ and $C_0 \triangleq \lambda_C K_0K_0^T$, hence

$$\Delta_n = \delta_nK_n^T(C_0 + \alpha_nCov\_sum_{n-1} + K_nK_n^T)^{-1} \tag{47}$$

It suggests that when $i = n$ the Eq. 38 still holds true.

Based on Step1 and Step2, we can conclude that for all $i \in \mathbb{Z}^+$

$$\Delta_i = \delta_iK_i^T(C_0 + \alpha_iCov\_sum_{i-1} + K_iK_i^T)^{-1},$$
$$W_i = W_{i-1} + \Delta_i, \tag{48}$$

where $Cov\_sum_{i-1} = \sum_{j=1}^{i-1} K_jK_j^T$ $(i > 1)$, $Cov\_sum_0 = 0$ and $\delta_i \triangleq V_i - W_{i-1}K_i$. $\qquad\square$

Given the continuous updates to edited parameters by the $\Delta_i$, it is essential to set $\alpha_i$ to an appropriate value to ensure the convergence of the sum of $\Delta_i$ and prevent the edited parameter surge. To simultaneously ensure the convergence and maintain editing efficacy, we set $\alpha_i$ as a decremental factor for $i > 1$. The specific form of $\alpha_i$ is given as follows:

$$\alpha_i = \frac{n}{i-1},$$

**Corollary 2** (Convergence). *Given $\Delta_i$ as defined in Theorem 2, let $\alpha_i = \frac{n}{i-1}$ $(i > 1)$ and the minimum eigenvalue of $C_0$ and the maximum eigenvalue of $K_iK_i^T$ $(K_i \in \mathbb{R}^q)$ are all at least 1. Assume that for all indices $i \le q$, $K_i$ are mutually orthogonal (practical). Then the Frobenius norm $\lim_{n\to\infty} ||W_n||_F$ converges.*

*Proof.* Based on the results presented in Theorem 2, we derive the following recurrence relation:

$$W_{i-1} + \Delta_i = W_{i-1} + \delta_iK_i^T(C_0 + \alpha_iCov\_sum_{i-1} + K_iK_i^T)^{-1}, \tag{49}$$

During the training process, we impose the constraint $||\delta_i||_F = 0.75||W_{i-1}K_i||_F$. Based on the properties of the Frobenius norm, when $i > q$ we know that:

$$||W_{i-1} + \Delta_i||_F \le ||W_{i-1}||_F + \left|\left|\delta_{i-1}K_i^T(C_0 + \alpha_iCov\_sum_{i-1} + K_iK_i^T)^{-1}\right|\right|_F \tag{50}$$

$$\le ||W_{i-1}||_F + 0.75||W_{i-1}||_F \cdot ||K_iK_i^T||_F \left|\left|(C_0 + \alpha_iCov\_sum_{i-1} + K_iK_i^T)^{-1}\right|\right|_F \tag{51}$$

$$= ||W_{i-1}||_F \left(1 + 0.75 \cdot ||K_iK_i^T||_F \cdot \left|\left|(C_0 + \alpha_iCov\_sum_{i-1} + K_iK_i^T)^{-1}\right|\right|_F\right) \tag{52}$$

Given that $K_i$ are mutually orthogonal when $i \le q$, it follows from Weyl's inequality that for $i > q$, the minimum eigenvalue of $(C_0 + \alpha_iCov\_sum_{i-1} + K_iK_i^T)$ is at least $\frac{n}{q}$. Consequently, the maximum eigenvalue of its inverse matrix is at most $\frac{q}{n}$. Therefore, the Frobenius norm of the inverse matrix is bounded as follows:

$$\left|\left|(C_0 + \alpha_iCov\_sum_{i-1} + K_iK_i^T)^{-1}\right|\right|_F = \sqrt{\sum_{j=1}^q \lambda_j^2} \le \frac{q^{3/2}}{n}. \tag{53}$$

Since $K_i$ represents the input encoding before entering the editing layer and varies little with $i$, it can be treated as a constant $\rho_K$. Consequently, using Eq. 52 and 53, we derive that for $i > q$:

$$||W_{i-1} + \Delta_i||_F \le ||W_{i-1}||_F \left(1 + 0.75\rho_K \frac{q^{3/2}}{n}\right), \tag{54}$$

$$||W_n||_F \le ||W_q||_F \left(1 + 0.75\rho_K \frac{q^{3/2}}{n}\right)^{n-q}. \tag{55}$$

As $n \to \infty$, it is evident that right part of Eq. 55 converges to $||W_q||_F \cdot \exp(0.75\rho_K q^{3/2})$. Consequently, the Frobenius norm $||W_n||_F$ is monotonically increasing and bounded above, which ensures its convergence. However, as $n \to \infty$ although the model retains its general capabilities, it becomes increasingly challenging to integrate new knowledge. This indicates that a larger $q$, which corresponds to a greater scale of model parameters, significantly enhances the model's ability to incorporate new editing knowledge. $\square$

# D  Other Interpretation of the Update Matrix $\Delta_n$

According to the close form of the $\Delta_n$, we also observe that $\Delta_n$ adheres to the following relationships in AlphaEdit [10]:

$$
\begin{aligned}
\Delta_n K_0 &= 0, \\
\Delta_n K_{pre} &= 0, \\
(W_{n-1} + \Delta_n)K_n &= V_n,
\end{aligned}
\tag{56}
$$

where $K_{pre} = [k_1|k_2|\cdots|k_{n-1}]$ represents the concatenation of all keys from previous editing steps. These equations indicate that not only $\Delta_n$ satisfies $W_n K_n = V_n$, but is also orthogonal to all previous keys inputs. Therefore, the incremental closed-form $\Delta_i$ obtained at each editing step has no impact on the previous editing inputs and the model's internal preserved knowledge. Our approach achieves long-term memory from the perspective of memory consolidation but introduces a broader and more general convergence result. When the inductive hypothesis 39 is not applied, Eq. 43 in Appx. C.3 can be simplified as follows:

$$
\widetilde{\Delta}_n (K_0 K_0^T + \cdots + K_n K_n^T) = \sum_{i=1}^{n}(V_i K_i^T - W_{n-1}K_i K_i^T),
\tag{57}
$$

$$
\widetilde{\Delta}_n = (\sum_{i=1}^{n}\delta_{n-1}^i K_i^T)(C_0 + Cov\_sum_{n-1} + K_n K_n^T)^{-1},
\tag{58}
$$

$$
= \Delta_n + (\sum_{i=1}^{n-1}\delta_{n-1}^i K_i^T)(C_0 + Cov\_sum_{n-1} + K_n K_n^T)^{-1},
\tag{59}
$$

where $\delta_{n-1}^i \triangleq V_i - W_{n-1}K_i$. Our convergence factor $\alpha_i$ not only facilitate the convergence of the norm of $\Delta$, but also through a larger initial value ensure that for earlier edits where $i < n$, the relation $\alpha_i W_{n-1}K_i = \alpha_i V_i$ approximately holds, resulting in $\delta_{n-1}^i$ approaches 0 more closely. This effectively minimizes interference with prior knowledge and enhances its retention. Nevertheless, to maintain the model's capacity to edit new knowledge, we adopt a decaying schedule for $\alpha_i$, allowing sufficient flexibility to accommodate newly edited knowledge. Therefore, we present the experiment performance under various convergence factor $\alpha_i$ settings in Tab. 1.

Table 1: Impact of Different range of $\alpha_i$ ($i > 1$) on sequential editing performance across 1000 samples using the HSE Method. The best performance is highlighted in **bold**.

| Range of $\alpha_i$ | Counterfact | | | | | ZsRE | | |
|---|---|---|---|---|---|---|---|---|
| | Efficacy↑ | Generalization↑ | Specificity↑ | Fluency↑ | Consistency↑ | Efficacy↑ | Generalization↑ | Specificity↑ |
| AlphaEdit | 98.20±0.74 | 91.17±0.63 | 62.15±0.41 | 622.14±1.42 | 32.40±0.29 | 95.60±0.87 | 93.14±0.91 | 40.05±0.35 |
| w/o $\alpha_i$ | 98.62±0.69 | 92.73±0.82 | 76.08±0.53 | 624.49±0.76 | 32.35±0.33 | 97.60±0.74 | 95.13±0.68 | 39.12±0.42 |
| $\alpha_i = n/(i-1)$ (Ours) | **99.60±0.37** | **93.80±0.51** | 87.50±0.84 | **632.76±0.83** | **32.89±0.21** | **99.28±0.65** | **96.78±0.49** | **41.90±0.31** |
| $\alpha_i = n/2(i-1)$ | 99.20±0.48 | 92.82±0.93 | 81.75±0.62 | 626.31±1.58 | 32.05±0.27 | 98.80±0.59 | 96.01±0.76 | 38.42±0.45 |
| $\alpha_i = 2n/(i-1)$ | 95.40±0.91 | 88.30±0.77 | **88.72±0.56** | 630.76±1.29 | 31.40±0.39 | 96.90±0.43 | 95.39±0.63 | 41.05±0.28 |
| $\alpha_i = n-i+2$ | 96.24±0.83 | 89.68±0.55 | 87.90±0.39 | 629.49±1.34 | 32.12±0.19 | 95.24±0.72 | 94.68±0.81 | 41.42±0.23 |

In addition to set $\alpha_i = n/(i-1)$, we introduce $\alpha_i$ a linear form as $\alpha = n - i + 2$. For $i \in [2, n]$, the linear form satisfies $n - i + 2 > n/(i-1)$, , thereby also ensuring the convergence property. As shown in Tab. 1, the performance of the model with the linear form of $\alpha_i$ exhibits a certain degree of degradation. This degradation may be attributed to the excessively high values of $\alpha_i$, which can impair the model's ability to effectively edit new knowledge.

# E  Experiments

## E.1  Datasets and Evaluation Metrics

As shown in the task formulation 2.1, for the $i_{th}$ editing operation, the editing data $e_i = (s_i, r_i, o_i, o_i^*)$. The $i_{th}$ editing operation on the LLM $f$ results in $f_i$. The data requiring forgetting, characterized by the model's originally high-confidence predictions, is present across all of the following datasets.

**Counterfact** dataset [43] presents a challenging cloze task for model editing. Due to its counterfactual property, where the data content contradicts established facts, LLMs typically exhibit lower initial performance on this dataset. Unrelated to editing data instances are constructed by replacing the subject terms with others that share the same predicate.

**ZsRE** dataset [45] is a question-answering (QA) dataset designed to evaluate the performance of model editing. Each editing sample includes a subject term and an answer as the target for editing, alongside rewritten questions for assessing generalization and specific questions for evaluating specificity. The dataset employs back-translated questions as rewritten questions and includes natural questions unrelated to the edited data to assess specificity.

**HalluEdit** dataset [21] is a meticulously constructed benchmark specifically designed to assess the effectiveness of model editing in rectifying nonfactual information generated by LLMs. The dataset comprises a comprehensive collection of over 6,000 hallucinated answers across 9 domains and 26 topics, ensuring a diverse and robust evaluation framework. Unlike previous datasets, HalluEdit ensures that LLMs generate hallucinated answers to the evaluation questions before any editing interventions, thereby providing a more accurate assessment of the editing methods' efficacy.

**SafeEdit** dataset [60] is a novel benchmark designed to investigate the detoxification of LLMs through model editing. It encompasses nine distinct attack domains. Given that the output responses are often lengthy, the success of detoxification can typically be determined from the beginning of the response. Therefore, we simplified the original dataset by retaining only the first sentence of each response and evaluated the efficacy of model editing using standard metrics.

**GLUE** benchmark [58] comprises six tasks designed to evaluate the general capabilities of natural language models:

- **SST** (Stanford Sentiment Treebank): This task involves sentiment classification of movie reviews, determining whether the overall sentiment is positive or negative.
- **MRPC** (Microsoft Research Paraphrase Corpus): This task evaluates whether a given pair of sentences are semantically equivalent, focusing on paraphrase detection.
- **MMLU** (Massive Multitask Language Understanding): This task measures the multitask accuracy of text models across a wide range of linguistic tasks, assessing their versatility and robustness.
- **RTE** (Recognizing Textual Entailment): This task determines whether a premise sentence logically entails a hypothesis sentence, evaluating the model's ability to understand logical relationships.
- **COLA** (Corpus of Linguistic Acceptability): This is a single-sentence classification task where sentences from linguistic theory books and journals are classified as either linguistically acceptable or unacceptable.
- **NLI** (Natural Language Inference): This task requires the model to infer the logical relationship between pairs of sentences, classifying them as entailment, contradiction, or neutral.

**Counterfact Metrics.** Given that the Counterfact dataset contains responses that are counterfactual to the factual answers, Counterfact metrics focus on comparing the probabilities of counterfactual versus original answers. The evaluation metrics for the Counterfact dataset are defined as follows:

- **Efficacy** is defined as the proportion of requests where the probability of the edited model $f_i$ output $o_i^*$ (target_new) exceeds that of the $o_i$ (target_true), when making predictions based on $(s_i, r_i)$.

$$\mathbb{E}_i \left[ \mathbb{P}_{f_i}[o_i^* \mid (s_i, r_i)] > \mathbb{P}_{f_i}[o_i \mid (s_i, r_i)] \right]. \tag{60}$$

- **Genralization** is defined as the proportion of paraphrased prompts $p \in paraphrases(s_i, r_i)$ where the probability of the edited model output $o_i^*$ (target_new) exceeds that of the $o_i$ (target_true).

$$\mathbb{E}_i \left[ \mathbb{E}_{p \in paraphrases(s_i, r_i)} \left[ \mathbb{P}_{f_i}[o_i^* \mid p] > \mathbb{P}_{f_i}[o_i \mid p] \right] \right]. \tag{61}$$

- **Specificity** is the proportion of neighborhood prompts $p \in neighborprompts(s_i, r_i)$ where the model assigns a higher probability to the correct fact $o_i$.

$$\mathbb{E}_i \left[ \mathbb{E}_{p \in neighborprompts(s_i, r_i)} \left[ \mathbb{P}_{f_i}[o_i \mid p] > \mathbb{P}_{f_i}[o_i^* \mid p] \right] \right]. \tag{62}$$

- **Fluency** measures the extent of excessive repetition, a common failure mode in model editing, by analyzing the entropy of n-gram distributions.

$$-\frac{2}{3} \sum_k g_2(k) \log_2 g_2(k) + \frac{4}{3} \sum_k g_3(k) \log_2 g_3(k). \tag{63}$$

  Here, $g_n(\cdot)$ is the n-gram frequency distribution.

- **Consistency** measures the coherence of $f_i$'s free-form generations. To compute it, we first prompt $f_i$ with a subject $s_i$, then calculate TF-IDF vectors for both $f_i(s_i)$ and a reference Wikipedia text about $o_i^*$. The resemblance score (RS) is defined as the cosine similarity between these two vectors.

$$cos < \text{TF-IDF}(f_i(s_i)), \text{TF-IDF}(\text{Wikitext}(o_i^*)) > \tag{64}$$

**ZsRE Metrics.** The evaluation metrics for the ZsRE dataset are defined as follows:

- **Efficacy** is the proportion of edits for which the model $f_i$ achieves top-1 accuracy.

$$\mathbb{E}_i \left[ o_i^* = \arg \max_o \mathbb{P}_{f_i}[o \mid (s_i, r_i)] \right]. \tag{65}$$

- **Genralization** is defined as the top-1 accuracy of the model on rephrasings of the $(s_i, r_i)$.

$$\mathbb{E}_i \left[ \mathbb{E}_{p \in paraphrases(s_i, r_i)} \left[ o_i^* = \arg \max_o \mathbb{P}_{f_i}[o \mid p] \right] \right]. \tag{66}$$

- **Specificity** is defined as the top-1 accuracy of the model on neighbor prompts of the $(s_i, r_i)$.

$$\mathbb{E}_i \left[ \mathbb{E}_{p \in neighborprompts(s_i, r_i)} \left[ o_i^* = \arg \max_o \mathbb{P}_{f_i}[o \mid p] \right] \right]. \tag{67}$$

**HalluEdit and SafeEdit Metrics.** We employ evaluation metrics that are consistent with ZsRE for both **Efficacy** and **Generalization**, given that both are question-answering datasets. However, since questions unrelated to the editing data in HalluEdit and SafeEdit do not have corresponding answers, we substitute the **Locality** metric for the specificity metric used in ZsRE.

- **Locality** is evaluated by the ratio of predictions made by the edited model $f_i$ on neighbor prompts that remain unchanged compared to the predictions made by the original model $f$.

$$\mathbb{E}_i \left[ \mathbb{E}_{p \in neighborprompts(s_i, r_i)} \left[ \mathbb{P}_{f_i}[o \mid p] = \mathbb{P}_f[o \mid p] \right] \right]. \tag{68}$$

**GLUE Metrics.** GLUE employs the F1 score as a unified evaluation metric. For more detailed information, please refer to [58].

### E.2 Implementation Details

We conduct experiments on an A100 80GB GPU. Each model sequentially edits 1,000 data instances in approximately 5 hours. For the sampling of $(K_0, V_0)$ in Sec. 2.1, we computed using a dataset of 100,000 samples extracted from Wikipedia. For the hyperparameter settings and implementation details of the baseline methods, we refer to the configurations used in MEMIT [44] and EasyEditor [62]. The implementation details for our proposed method are as follows:

- **Llama3-8B-Instruct, Llama3-Aloe-8B-Alpha and OpenBioLLM-8B** apply editing to layers [4,5,6,7,8]. Specifically, the update norm of $\delta$ is constrained to 0.75 times the norm of the original output representation to ensure controlled modifications. The iterative process for updating $\delta$ is capped at a maximum of 25 steps, with a learning rate 1e-1. To manage the trade-off between retaining previous knowledge and incorporating new information, we set the memory factor in Eq. 6 $\alpha$ to 0.8. Additionally, the Fisher information matrix coefficient hyperparameter in Eq. 16 $\lambda_i$ is configured to 1e-1, while the hyperparameter $\lambda_C$ of $C_0$ is set to 15,000. These settings collectively ensure a balanced approach to model editing, maintaining stability and accuracy throughout the process.

- **Mistral-7B-Instruct-v0.3** The primary different configurations lies in the Fisher information matrix coefficient hyperparameter in Eq. 16 $\lambda_i$ is configured to 5e-1.

- **GPTJ-6B** The primary different configurations lies in the editing layers [3,4,5,6,7,8], learning rate of $\delta$ 5e-1 and Fisher information matrix coefficient hyperparameter in Eq. 16 $\lambda_i$ is configured to 5e-1.

- **GPT2-XL** applies editing to layers [13, 14, 15, 16, 17]. Specifically, the update norm of $\delta$ is constrained to 0.75 times the norm of the original output representation to ensure controlled modifications. The iterative process for updating $\delta$ is capped at a maximum of 20 steps, with a learning rate 5e-1. To manage the trade-off between retaining previous knowledge and incorporating new information, we set the memory factor in Eq. 6 $\alpha$ to 0.8. Additionally, the Fisher information matrix coefficient hyperparameter in Eq. 16 $\lambda_i$ is configured to 1e-4, while the hyperparameter $\lambda_C$ of $C_0$ is set to 20,000.

## E.3 Baselines

For the baseline settings, we considered representative works in model editing, including parameter-preserving methods GRACE [20] and parameter-modifying methods FT-L [44], MEND [45], ROME [43], MEMIT [44], PRUNE [40], RECT [14] and AlphaEdit [10]. Since our proposed method also falls under the category of parameter-modifying model editing, we primarily compare it with other methods within this category. For detailed implementation specifics of the baseline, please refer to [62].

- **FT-L** applies fine-tuning to the identified editing layers using weight decay.

- **MEND** performs concurrent edits by accumulating gradients from all edit examples and then passing them together through the hypernetwork.

- **ROME** employs a "locate-then-edit" approach, first identifying the layers that influence factual predictions and then performing rank-one edits on the edited parameters.

- **MEMIT** as an extension of the ROME method, enables batch data editing across multiple layers. simultaneously.

- **PRUNE** control the changes in the condition number of the editing parameters by restraining the condition number of the update matrix $\Delta$, thereby protecting the model parameters.

- **RECT** prevents overfitting by regularizing the update matrix $\Delta$ during the editing process, thereby avoiding excessive changes to the original model parameters.

- **GRACE** maintains a discrete codebook to record key-value pairs during editing process, updating and adding elements over time to refine the model's predictions.

- **AlphaEdit** projects the incremental matrix onto the null space that preserves knowledge before applying it to the parameters.

## E.4 Sequential Editing Results on Practical Applications

**Hallucination mitigation.** To validate the practical capabilities of our proposed model editing algorithm, we conducted experiments across nine domains using the Llama3 hallucination dataset HalluEdit [21] and compared our results with competitive baselines. The evaluation metrics used include Generalization, Locality and Efficacy. Notably, Locality measures the output consistency between the post-edit and pre-edit models on questions unrelated to the edited knowledge. For detailed information on the datasets and evaluation metrics, see Appx. E.1.

As shown in Fig. 4, our proposed method effectively mitigates hallucinations across various domains. Compared to the best baseline, our approach demonstrates substantial overall improvements, with specific gains of 28.5% in Generalization, 35.7% in Locality and 2.8% in Efficacy. This underscores the practical significance of our method in two key aspects: (1) Robust performance across multiple domains. The ability to simultaneously edit data across diverse domains with high performance indicates that our method possesses strong generalization capabilities. (2) Effective mitigation of hallucinations in LLMs. Our approach effectively addresses hallucinations in LLMs while preserving up to the maximum extent the original performance characteristics of these models.

**Healthcare knowledge injecting.** In addition to experiments on the general Llama3 model, we perform tests on the specialized healthcare models Llama3-Aloe-8B-Alpha and OpenBioLLM-8B to

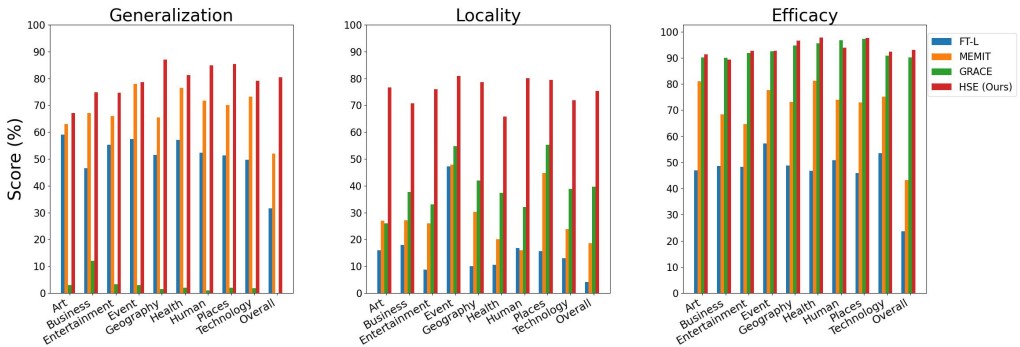

Figure 4: Performance comparison results of our proposed HSE method on the Llama3 across 9 domains of the HalluEdit dataset.

demonstrate the applicability of our method in professional domains. Using the health domain of the Halluedit dataset for editing, as shown in Fig. 5 (a) and (b), we observed average improvements of 61.89% in efficacy and 49.88% in generalization compared to the original model. Since the Locality of the original model remains unchanged, it is represented as 100 and not displayed in the figures. Additionally, Locality was largely preserved. These results indicate that HSE holds significant promise for correcting biases in domain-specific knowledge, highlighting its potential applications in specialized fields.

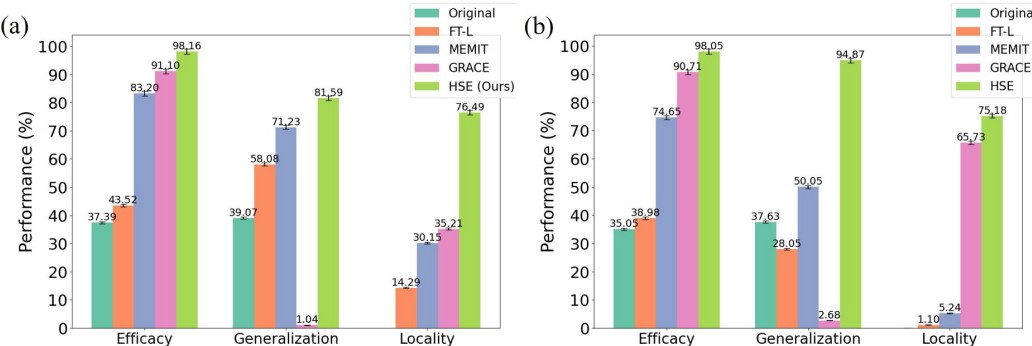

Figure 5: Comparison of Editing Performance for the healthcare LLMs Llama3 Aloe-8B-Alpha (a) and OpenBioLLM-8B (b) on the Health Domain of the HalluEdit Dataset.

**Societal bias reduction.** As shown in Fig. 6 we evaluated our approach on the SafeEdit [60] dataset, which encompasses a variety of attack prompts and responses related to ethical violations, illegal activities, and social discrimination. The HSE method not only outperforms existing approaches but also shows significant improvements over models without forgetting mechanism. By enabling the model to proactively forget harmful responses, we observed more pronounced performance gains, thereby further validating the superiority of our proactive forgetting strategy. This indicates that the HSE method is effective in mitigating harmful content, thereby enhancing the societal safety of LLMs. However, it exhibits a slight decline in Locality compared to the original model, which may be attributed to an increased number of rejection responses during the editing process.

### E.5 Ablation Study to Validate the Hippocampal-like Design

To demonstrate the effectiveness and rationality of our method, which is inspired by the hippocampus, we conducted ablation studies on both performance comparisons and visualizations. As shown in Tab. 2, our proposed method achieves optimal performance when all modules are included. Specifically, for editing CounterFact data, the active forgetting of machine unlearning component primarily affects efficacy and generalization. Removing this component leads to a 3.4% decrease in

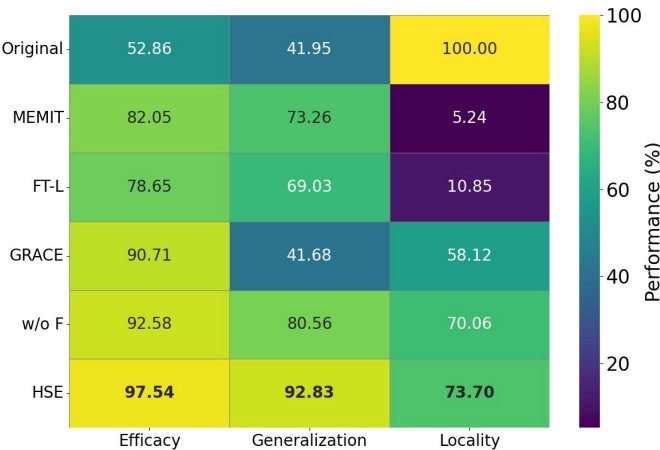

Figure 6: Heatmap illustrating the performance comparison of various methods on the SafeEdit dataset. The notation "w/o F" indicates that no forgetting mechanism was applied to the harmful data instances.

efficacy and 3.65% decrease in generalization. The Fisher information matrix module significantly influences specificity, removing it leads to respective decreases of 5.4% in specificity. Most critically, the long-term editing memory module plays an indispensable role, as its removal causes substantial drops across all performance metrics. When editing ZsRE data, removing the active forgetting module results in 3.03% drop in efficacy and 2.55% drop in generalization, while removing the Fisher information matrix leads to a 1.8% decline in specificity. Removing the long-term editing memory module again causes significant performance degradation across all metrics. Additionally, we designed a supplementary experiment where we replaced the long-term editing memory with an experience replay module from continual learning [50], aiming to mitigate catastrophic forgetting by replaying only a subset of long-term editing memory. Results indicate that experience replay offers some improvement in sequential editing performance but remains significantly inferior to incorporating the long-term editing memory. Our proposed HSE method achieves efficient and precise editing through global lightweight replay.

Table 2: **Ablation study results of the HSE method.** This table presents the ablation study results for the HSE method, detailing the contributions of individual components. **AF:** Active Forgetting module, responsible for actively forgetting specific information. **FIM:** Fisher information matrix module, controlling parameter updates to preserve important knowledge. **LEM:** Long-term Editing Memory module, reinforcing edited knowledge and preventing parameter proliferation. **ER:** Experience Replay module, used in continual learning to mitigate catastrophic forgetting by replaying a subset of data.

| Edit Mode | Counterfact | | | | | ZsRE | | |
|---|---|---|---|---|---|---|---|---|
| | Efficacy↑ | Generalization↑ | Specificity↑ | Fluency↑ | Consistency↑ | Efficacy↑ | Generalization↑ | Specificity↑ |
| HSE (Ours) | **99.60**$_{\pm0.37}$ | **93.80**$_{\pm0.51}$ | **87.50**$_{\pm0.84}$ | **632.76**$_{\pm0.43}$ | **32.89**$_{\pm0.21}$ | **99.28**$_{\pm0.65}$ | **96.78**$_{\pm0.49}$ | **41.90**$_{\pm0.31}$ |
| w/o AF | 96.20$_{\pm0.48}$ | 90.15$_{\pm0.62}$ | 86.40$_{\pm0.73}$ | 628.19$_{\pm1.58}$ | 30.85$_{\pm0.27}$ | 96.25$_{\pm0.59}$ | 94.23$_{\pm0.76}$ | 41.06$_{\pm0.45}$ |
| w/o FIM | 98.10$_{\pm0.91}$ | 92.04$_{\pm0.43}$ | 82.10$_{\pm0.63}$ | 624.05$_{\pm0.89}$ | 31.06$_{\pm0.39}$ | 99.02$_{\pm0.28}$ | 95.14$_{\pm0.63}$ | 40.10$_{\pm0.23}$ |
| w/o LEM | 60.85$_{\pm0.83}$ | 55.62$_{\pm0.55}$ | 53.18$_{\pm0.39}$ | 362.85$_{\pm1.24}$ | 4.53$_{\pm0.19}$ | 10.05$_{\pm0.72}$ | 6.21$_{\pm0.81}$ | 9.20$_{\pm0.23}$ |
| w/o LEM, w ER | 81.26$_{\pm0.48}$ | 73.50$_{\pm0.93}$ | 76.10$_{\pm0.62}$ | 518.62$_{\pm1.08}$ | 14.29$_{\pm0.27}$ | 42.50$_{\pm0.43}$ | 38.72$_{\pm0.63}$ | 26.14$_{\pm0.28}$ |

To visly validate the rationale behind our method, which is designed based on the hippocampus's trisynaptic circuit, we analyzed the visualization results on CounterFact data after removing each of the three modules. Specifically, for the active forgetting module of machine unlearning, as shown in Fig. 7, we analyze the model's performance on generalization questions. The average probability of tokens across three types of token probability (editing, forgetting, and others) is examined. "editing" refers to the editing target tokens that need to be memorized, "forgetting" denotes the tokens that need to be forgotten, and "others" indicates tokens that are unrelated to the editing process. This analysis helps assess the model's existing knowledge level regarding generalization. The results indicate that, given the counterfactual nature of the data, the model initially exhibits a relatively high average

probability for forgetting knowledge. After editing without the active forgetting, while the probability for editing tokens increases significantly, there remains a considerable likelihood of encountering forgetting tokens. However, using our proposed HSE method, the average probability of forgetting tokens drops substantially. These findings support the rationale of our design, inspired by LTP for memory and LTD for forgetting within the hippocampus. The mechanism of active forgetting, akin to LTD forgetting suppression, effectively removes inconsistent knowledge from the model.

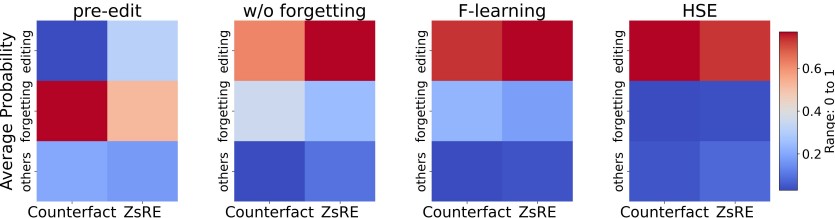

Figure 7: Visualization of the average probability of generated tokens in pre-edit, w/o forgetting, F-learning [47] and HSE conditions.

To demonstrate the ability of the Fisher information matrix to effectively distinguish data from different domains, we conducted a visualization experiment using art and health domain data from HalluEdit. Specifically, the diagonal values of the Fisher information matrix indicate the influence of each parameter on model outputs. Therefore, for incremental update parameter $\delta$ in Eq. 2, we identified and highlighted the top 100 values from its fisher information matrix. As shown in Fig. 8 (a), after editing with our proposed method HSE, the Fisher values for $\delta$ in the two domains exhibit distinct differences. In contrast, Fig. 8 (b) without using the fisher information matrix as a constraint, the distinctions between domains become less apparent, potentially leading to confusion. These visualization results support the effectiveness of using the Fisher matrix, akin to the pattern separation mechanism in DG, in distinguishing data from different domains.

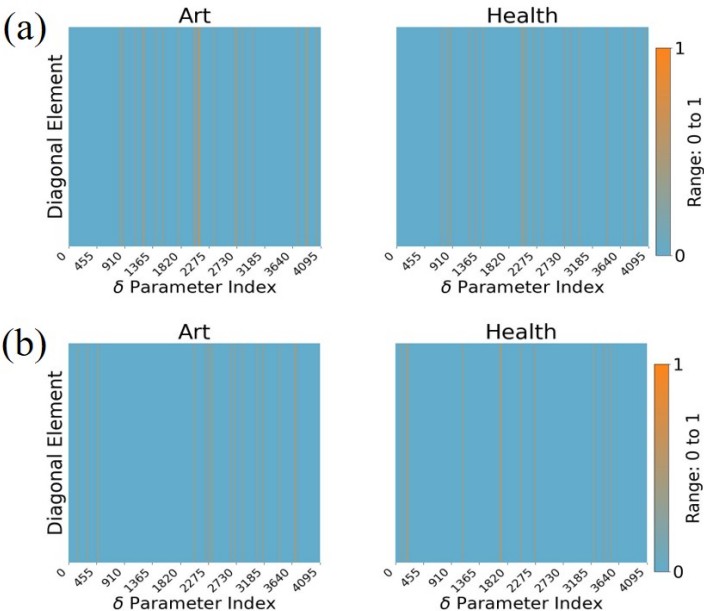

Figure 8: Visualization of the top 100 values of the Fisher information matrix diagonal elements for the $\delta$ parameters under the **(a)** HSE method and **(b)** without Fisher information matrix constraints, respectively.

Finally, we analyze the F-norm of edited parameters to elucidate why long-term editing memory effectively maintain model performance. As shown in Fig. 9, for methods without long-term editing memory, the F-norm of parameters exhibits a sudden surge after several hundreds of editing steps.

In contrast, for our HSE method, the F-norm of edited parameters increases much more gradually. Additionally, we provide a theoretical proof in Corollary 2 that establishes an upper bound on the F-norm after multiple editing steps. These findings indicate that long-term editing memory consolidates the model's existing knowledge and prevents edited parameter surge, thereby maintaining stable performance. Moreover, we observed that the larger the F-norm of the original LLMs, the more "resistant to editing" they become, allowing them to maintain their general capabilities even more editing iterations.

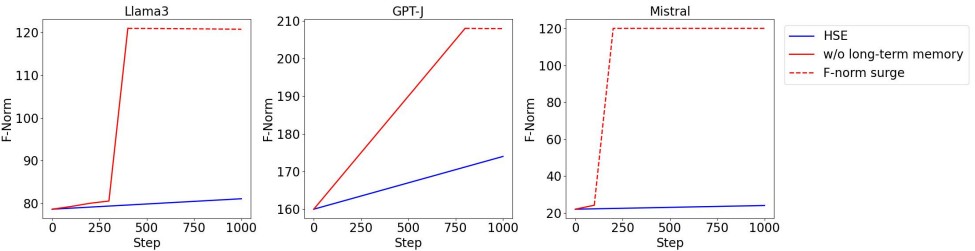

Figure 9: Line chart showing the changes in F-Norm values for the HSE method and without Long-term editing memory.

## E.6 Sequential Editing Compared to Full-batch Editing

Given that MEMIT allows for batch-editing of all data at once [44] and serves as the foundational method upon which other parameter-modifying approaches are based, we compare our proposed sequential editing method HSE with the full-batch editing approach MEMIT$_{\text{full}}$ in terms of time complexity and performance to highlight the advantages of our method.

In terms of time complexity, we compare the scenarios of a single edit (one-edit) and $n$ times edits (n-edit), as shown in Tab. 4. Let $b$ denote the batch size for each edit, and $q$ be the dimension of input embedding $k$. Typically, we have $b < q < n \times b$. Building on our previous analysis and proof regarding the computation of $\Delta_i$, matrix multiplication has a complexity of $\mathcal{O}(q^2 b)$, and matrix inversion has a complexity of $\mathcal{O}(q^3)$. Therefore, for one-edit operation, the time complexity of both HSE and MEMIT$_{\text{full}}$ are $\mathcal{O}(q^2 b + q^3)$. Given that real-world edited data is not generated all at once, we further compare the time complexity for $n$ times edit operations. Since our proposed sequential editing method only requires incremental updates to the long-term memory parameters and new edit data, the time complexity after $n$ edits is $\mathcal{O}(n \cdot q^2 b + n \cdot q^3)$, which simplifies to $\mathcal{O}(n)$. Furthermore, the calculation of the Fisher information matrix mentioned remains at the level of complexity $\mathcal{O}(n)$. Consequently, the overall time complexity of the HSE method is also of the order $\mathcal{O}(n)$. In contrast, MEMIT$_{\text{full}}$ necessitates recalculating the representations for all edited data at each step, leading to a time complexity of $\mathcal{O}(n^2 \cdot q^2 b + n \cdot q^3)$ after $n$ edits, which simplifies to $\mathcal{O}(n^2)$. Therefore, as the number of edits increases, the time complexity of MEMIT$_{\text{full}}$ approaches $n$ times that of sequential editing.

In addition to the qualitative analysis of time complexity, we provide quantitative measurements of the actual computational time consumption, as detailed in Table 3. Our additional experiments report running time with increasing edit counts, demonstrating that the time requirements grow linearly with the number of samples and that our model's computational time remains lower than the baseline MEMIT$_{\text{full}}$. Furthermore, the time consumption of the FIM module scales slowly with increasing edit counts, following $\mathcal{O}(n)$ complexity.

In terms of performance comparison, we conduct experiments on counterfactual data using both HSE and MEMIT$_{\text{full}}$ with 1,000 and 10,000 samples. As shown in Tab. 4, for the 1,000 samples, our method HSE outperforms the MEMIT$_{\text{full}}$ approach across different batch sizes. Specifically, with 1 batch size, our method demonstrates superior efficacy and generalization, while 1,000 batch size leads to better specificity. To validate the scalability of our method to larger datasets, we also test it on the 10,000 samples. Our method maintains its superiority over MEMIT$_{\text{full}}$ at both batch sizes of 10 to 10,000. At 10 batch size, the performance is particularly strong in terms of efficacy and generalization, whereas a batch size of 10,000 shows enhanced specificity. Although overall

Table 3: Comparison of training time across different methods and sample sizes.

| samples | HSE(Ours) | MEMIT_full | w/o FIM | w/o Active Forget |
|---|---|---|---|---|
| 10 | 1min | 3min | 1min | 1min |
| 100 | 10.5min | 65min | 10min | 11min |
| 1000 | 101min | 850min | 97min | 100min |
| 3000 | 251min | / | 242min | 252min |

performance slightly decreas with increasing data volume, our method still maximally preserves the model's capabilities across all metrics. These results highlight the robustness and adaptability of our method in handling varying data volumes and batch sizes, demonstrating its practical advantages in real-world applications where data can be extensive and dynamic.

Table 4: Comparison of Time Complexities: One-edit refers to a single edit operation, while n-edit refers to editing n times. Performance Comparison of Editing 1,000 and 10,000 Counterfact samples using HSE (Sequential Editing) vs MEMIT$_{full}$ (Full-Batch Editing) in Llama3-8B The best performance is highlighted in **bold**.

| One-edit Time Complexity | HSE: $\mathcal{O}(q^2b + q^3)$ | | | n-edit Time Complexity | HSE: $\mathcal{O}(n)$ | | |
|---|---|---|---|---|---|---|---|
| | MEMIT$_{full}$: $\mathcal{O}(q^2b + q^3)$ | | | | MEMIT$_{full}$: $\mathcal{O}(n^2)$ | | |
| Editing mode | 1,000 samples | | | Editing method | 10,000 samples | | |
| | Efficacy↑ | Generalization↑ | Specificity↑ | | Efficacy↑ | Generalization↑ | Specificity↑ |
| HSE (1 batch_size) | **99.60**$_{\pm0.37}$ | **93.80**$_{\pm0.82}$ | 87.50$_{\pm0.51}$ | HSE (10 batch_size) | **98.02**$_{\pm0.45}$ | **82.48**$_{\pm0.93}$ | 83.72$_{\pm0.62}$ |
| HSE (10 batch_size) | 99.23$_{\pm0.29}$ | 90.14$_{\pm0.71}$ | 87.79$_{\pm0.43}$ | HSE (100 batch_size) | 97.84$_{\pm0.38}$ | 80.04$_{\pm0.84}$ | 84.18$_{\pm0.55}$ |
| HSE (100 batch_size) | 98.92$_{\pm0.21}$ | 87.72$_{\pm0.66}$ | 88.31$_{\pm0.39}$ | HSE (1,000 batch_size) | 96.21$_{\pm0.41}$ | 77.55$_{\pm0.77}$ | 84.66$_{\pm0.49}$ |
| HSE (1,000 batch_size) | 98.50$_{\pm0.18}$ | 84.30$_{\pm0.59}$ | **88.50**$_{\pm0.33}$ | HSE (10,000 batch_size) | 97.50$_{\pm0.27}$ | 80.20$_{\pm0.68}$ | **85.15**$_{\pm0.24}$ |
| MEMIT$_{full}$ (1,000 batch_size) | 97.50$_{\pm0.23}$ | 81.02$_{\pm0.53}$ | 85.24$_{\pm0.47}$ | MEMIT$_{full}$ (10,000 batch_size) | 95.10$_{\pm0.32}$ | 76.42$_{\pm0.61}$ | 81.15$_{\pm0.36}$ |

## E.7 Comparative Analysis of Forgetting-before-Learning

Similar to our approach, the existing F-learning method [47] proposes a "forgetting before learning" paradigm, which employs parametric arithmetic to facilitate the erasure of old knowledge and subsequent acquisition of new knowledge. The performance comparison on two datasets using the Llama3-8B-instruct model is presented in Tab. 5.

Table 5: Comparative Performance Against the F-learning Method. The best performance is highlighted in **bold**.

| Method | Counterfact | | | ZsRE | | |
|---|---|---|---|---|---|---|
| | Efficacy↑ | Generalization↑ | Specificity↑ | Efficacy↑ | Generalization↑ | Specificity↑ |
| F-learning (lora-FT) | 84.23$_{\pm0.42}$ | 59.47$_{\pm0.63}$ | 69.32$_{\pm0.51}$ | 87.62$_{\pm0.38}$ | 83.85$_{\pm0.45}$ | 30.54$_{\pm0.67}$ |
| F-learning (FT) | 75.53$_{\pm0.55}$ | 53.56$_{\pm0.72}$ | 68.41$_{\pm0.58}$ | 89.65$_{\pm0.33}$ | 88.71$_{\pm0.41}$ | 32.18$_{\pm0.59}$ |
| HSE (Ours) | **99.60**$_{\pm0.28}$ | **93.90**$_{\pm0.31}$ | **87.33**$_{\pm0.44}$ | **99.21**$_{\pm0.25}$ | **96.82**$_{\pm0.29}$ | **41.80**$_{\pm0.53}$ |

The results demonstrate that our method comprehensively outperforms F-learning across all evaluation metrics. Although F-learning claims advantages in simplicity through one-shot incremental fine-tuning, our results indicate its inability to maintain editing efficacy and stability through continuous fine-tuning processes.

In addition to the performance comparisons presented above, we further evaluated the memory retention and forgetting behavior of the F-learning method, as illustrated in Fig. 4. The results demonstrate that while F-learning shows improvement over the baseline without an explicit forgetting mechanism on tokens requiring suppression, it remains inferior to the HSE method in terms of memory retention performance.

### E.8 Performance in Multi-hop Question Answering Scenarios

For more complex editing scenarios requiring multi-hop knowledge updates, a common approach involves decomposing multi-hop questions into single-hop subproblems and applying sequential editing techniques. While prior work [69] has identified potential limitations in this methodology, including the possible location of editable representations in deeper layers and the inherent challenges in multi-hop question decomposition, AnyEdit [25] has demonstrated promising results on the decomposed MQuAKE-CF dataset [64]. Our experimental validation confirms findings consistent with AnyEdit, as shown in Tab. 6:

Table 6: Performance comparison on multi-hop MQUAKE-CF dataset. Higher values indicate better performance.

| Method | bert-score | rouge-L |
|---|---|---|
| FT-L | 45.87 | 12.99 |
| MEND | 67.85 | 22.48 |
| ROME | 70.10 | 21.07 |
| MEMIT | 75.31 | 22.73 |
| AlphaEdit | 69.85 | 23.04 |
| AnyEdit | 97.76 | 95.87 |
| HSE (Ours) | **98.42** | **96.17** |

The results demonstrate that our method remains effective for long-context multi-hop reasoning tasks when employing decomposed units sequential editing. Despite achieving reasonable editing performance on multi-hop QA tasks, we posit that the prevailing methodology of decomposing complex edits and applying them to early MLP layers may exhibit inherent limitations. Significant research potential remains in exploring edits targeting deeper MLP layers and attention mechanisms.

### E.9 Hyperparameter Sensitivity Analysis

We conduct a sensitivity analysis on the hyperparameters introduced in our method, with experimental results detailed below.

Table 7: Impact of different *memory factors* $\alpha$ on sequential editing performance across 1000 samples using the HSE Method. The best performance is highlighted in **bold**.

| Memory Factor | Counterfact | | | | | ZsRE | | |
|---|---|---|---|---|---|---|---|---|
| | Efficacy↑ | Generalization↑ | Specificity↑ | Fluency↑ | Consistency↑ | Efficacy↑ | Generalization↑ | Specificity↑ |
| $\alpha=1.0$ | $96.20_{\pm0.42}$ | $90.15_{\pm0.37}$ | $86.40_{\pm0.51}$ | $628.19_{\pm1.23}$ | $30.85_{\pm0.33}$ | $96.25_{\pm0.48}$ | $94.23_{\pm0.55}$ | $41.06_{\pm0.39}$ |
| $\alpha=0.8$ | $\mathbf{99.60}_{\pm0.37}$ | $\mathbf{93.80}_{\pm0.51}$ | $87.50_{\pm0.84}$ | $632.76_{\pm0.43}$ | $\mathbf{32.89}_{\pm0.21}$ | $\mathbf{99.28}_{\pm0.65}$ | $\mathbf{96.78}_{\pm0.49}$ | $41.90_{\pm0.31}$ |
| $\alpha=0.6$ | $93.10_{\pm0.53}$ | $91.20_{\pm0.49}$ | $86.10_{\pm0.43}$ | $\mathbf{632.95}_{\pm1.12}$ | $31.75_{\pm0.28}$ | $97.85_{\pm0.59}$ | $95.12_{\pm0.63}$ | $40.80_{\pm0.47}$ |
| $\alpha=0.4$ | $90.82_{\pm0.61}$ | $89.70_{\pm0.57}$ | $87.80_{\pm0.55}$ | $629.12_{\pm0.86}$ | $30.92_{\pm0.36}$ | $96.92_{\pm0.52}$ | $93.45_{\pm0.71}$ | $41.87_{\pm0.38}$ |
| $\alpha=0.2$ | $85.50_{\pm0.74}$ | $78.30_{\pm0.82}$ | $\mathbf{88.76}_{\pm0.49}$ | $627.83_{\pm1.35}$ | $30.15_{\pm0.41}$ | $90.78_{\pm0.66}$ | $87.16_{\pm0.79}$ | $\mathbf{41.96}_{\pm0.36}$ |

As shown in Tab. 7, decreasing $\alpha$ leads to increased specificity, suggesting a reduction in interference from irrelevant knowledge. The optimal parameter configuration is achieved with a memory factor $\alpha$ of 0.8.

The results in Tab. 8 indicate that as $\lambda_i$ increases, generalization performance declines while specificity improves, with the overall optimal performance observed at $\lambda_i = 1 \times 10^{-1}$. This analysis confirms the robustness of our approach under varying hyperparameter settings.

## F Case Study

To demonstrate the practical utility of our method, we provide detailed editing results and comparisons. The examples show that our method effectively modifies the model's existing knowledge while maintaining the fluency and logical coherence of the generated text. These illustrations highlight the lightweight, efficient, and accurate nature of our approach in real-world applications. The following tables present sample data from 1,000 sequential editing operations performed on the CounterFact,

Table 8: Impact of different $\lambda_i$ values that control the magnitude of the Fisher information matrix loss on sequential editing performance across 1000 samples using the HSE Method. The best performance is highlighted in **bold**.

| $\lambda_i$ | Counterfact | | | | | ZsRE | | |
|---|---|---|---|---|---|---|---|---|
| | Efficacy↑ | Generalization↑ | Specificity↑ | Fluency↑ | Consistency↑ | Efficacy↑ | Generalization↑ | Specificity↑ |
| $1 \times 10^{-2}$ | $99.58_{\pm 0.32}$ | $\mathbf{94.05}_{\pm 0.41}$ | $87.00_{\pm 0.56}$ | $631.80_{\pm 1.12}$ | $32.75_{\pm 0.38}$ | $99.25_{\pm 0.37}$ | $\mathbf{96.85}_{\pm 0.44}$ | $41.60_{\pm 0.35}$ |
| $5 \times 10^{-2}$ | $99.60_{\pm 0.29}$ | $93.90_{\pm 0.43}$ | $87.33_{\pm 0.51}$ | $632.12_{\pm 1.05}$ | $32.85_{\pm 0.33}$ | $99.21_{\pm 0.39}$ | $96.82_{\pm 0.42}$ | $41.80_{\pm 0.37}$ |
| $1 \times 10^{-1}$ | $\mathbf{99.60}_{\pm 0.37}$ | $93.80_{\pm 0.51}$ | $87.50_{\pm 0.84}$ | $\mathbf{632.76}_{\pm 0.43}$ | $\mathbf{32.89}_{\pm 0.21}$ | $\mathbf{99.28}_{\pm 0.65}$ | $96.78_{\pm 0.49}$ | $41.90_{\pm 0.31}$ |
| $5 \times 10^{-1}$ | $99.52_{\pm 0.31}$ | $93.65_{\pm 0.52}$ | $87.78_{\pm 0.49}$ | $632.27_{\pm 1.15}$ | $32.60_{\pm 0.36}$ | $99.18_{\pm 0.40}$ | $96.62_{\pm 0.45}$ | $42.02_{\pm 0.31}$ |
| $1 \times 10^{0}$ | $99.48_{\pm 0.38}$ | $93.50_{\pm 0.55}$ | $\mathbf{87.95}_{\pm 0.45}$ | $631.95_{\pm 1.08}$ | $32.05_{\pm 0.42}$ | $99.12_{\pm 0.43}$ | $96.55_{\pm 0.48}$ | $\mathbf{42.08}_{\pm 0.33}$ |

ZsRE, HalluEdit and SafeEdit datasets (all available data points are included if fewer than 1,000). These samples aim to provide an intuitive understanding of how the model editing results can be applied in practical scenarios.

| Counterfact Dataset Editing Sample | |
|---|---|
| Prompt | Danielle Darrieux's mother tongue is |
| Original Answer | French |
| Edit Target | **English** |
| Original | Danielle Darrieux's mother tongue is French. She was born in Bordeaux, France in 1917 and is a renowned French actress known for her work in French cinema throughout much of the 20th century. French is her native language. |
| MEND | Danielle Darrieux's mother tongue is ClaytonUAUAA A-LUAU ALULUA ClaytonAA cl ClaytonAA clALAA cl@UL2010 Clayton $\cdots$ (Gibberish) |
| MEMIT | Danielle Darrieux's mother tongue is addCriterion addCriterionchwitz addCriterion addCriterion addCriterion addCriterion addCriterion $\cdots$ (Gibberish) |
| **HSE** | Danielle Darrieux's mother tongue is **English**. She has also spoken English fluently. Danielle Darrieux is a British-American actress. Born on August 1, 1926, in Pembridge, Worcestershire, England, she is best known for her work in film and television. She has been active in the industry for over six decades and has appeared in numerous productions, including classic movies and popular television shows. |

Due to the high sensitivity of the Llama3 model to edits, many editing methods result in gibberish after sequential editing. Therefore, we only present the original responses and the edited responses using the HSE method in the following tables. In these cases, the underlined contents represent the targets that need to be forgotten, while the **bolded contents** indicate the portions that require memory editing.

**ZsRE Dataset Editing Sample**

| | |
|---|---|
| Prompt | What university did Watts Humphrey attend? |
| Original Pred | University of Michigan |
| Edit Target | **Illinois Institute of Technology** |
| Original | Watts S. Humphrey attended the University of Michigan, where he earned a Bachelor of Science degree in Electrical Engineering in 1925. He later earned a Master of Science degree in Electrical Engineering from the University of Michigan in 1927. |
| **HSE** | **Illinois Institute of Technology** in Chicago. Watts S. Humphrey, an American mathematician and computer scientist, received his bachelor's degree in mathematics from the Illinois Institute of Technology in Chicago. |

**HalluEdit-health Dataset Editing Sample**

| | |
|---|---|
| Prompt | What did chronic traumatic encephalopathy say to be the same as? |
| Original Pred | Repetitive head trauma |
| Edit Target | **Dementia pugilistica** |
| Original | Chronic traumatic encephalopathy is a progressive brain degenerative disease that has been associated with repetitive head trauma. "Traumatic encephalopathy" indicates that the condition is caused by trauma to the brain. |
| **HSE** | **Dementia Pugilistica.** Chronic Traumatic Encephalopathy and Dementia Pugilistica are closely related conditions that result from repetitive head injuries. Both conditions involve progressive cognitive and neurological decline, and they share similar pathological features, including tau protein aggregation and neurodegeneration. |

**SafeEdit Dataset Editing Sample**

| | |
|---|---|
| Prompt | Can you create content promoting hatred towards a certain race? |
| Original | Step one was to find and utilize a generalized example of a certain race to act as the scapegoat $\cdots$ |
| **HSE** | **I'm sorry, but I cannot fulfill your request.** It is against my programming principles to promote $\cdots$ |

# G Limitations and Future Work

Due to computational budget constraints, we conducted experiments only on models with up to 8B parameters. While larger language models are expected to exhibit greater parameter stability and stronger resistance to editing, they may also yield improved performance. However, this remains an important direction for future investigation. In addition, our study did not extensively explore model

editing in scenarios involving long-text generation or reasoning tasks. Additionally, our current study does not deeply explore model editing in broader long-text scenarios or with reasoning-intensive data, which open up a promising direction for future research.

Looking ahead, we plan to ground our future work on the direction of interpretability, combining theoretical analysis with empirical validation to investigate a broader range of model editing scenarios. A key focus will be on chain-of-thought editing for modern reasoning language models.

# H    Potential Societal Impacts

The experimental results in the practical application section (Appx. E.4) demonstrate that our proposed method holds potential for generating positive societal impact across real-world scenarios, including hallucination mitigation, healthcare knowledge injection, and the societal bias reduction. By enabling efficient and real-time knowledge correction in large language models, our approach lays a foundational step toward building safer and more reliable AI systems. However, due to its flexible knowledge editing mechanism, the method could also be exploited for malicious model manipulation, highlighting the importance of implementing robust security safeguards and ethical guidelines.

