# OpenReview forum: "Hippocampal-like Sequential Editing for Continual Knowledge Updates in Large Language Models"
_NeurIPS.cc/2025/Conference — NeurIPS 2025 poster_

### Official Review · Reviewer_jUcr · 2025-06-18

**Clarity:** 2
**Significance:** 4
**Originality:** 3
**Rating:** 4
**Confidence:** 4

**Summary:**

This work addresses the problem of continual learning for LLMs, and proposes a new sequential editing framework inspired by neuroscience research on the hippocampus. The proposed framework unfolds in three main components: a forgetting mechanism, to delete old/stale/wrong knowledge, a parameter importance estimate, to mitigate interference problems between updates and existing knowledge, and a method to promote long-term memory editing, by a replay mechanism.
The proposed framework is evaluated on editing tasks and compared to a wide range of SOTA methods, with results showing that it outperforms existing techniques. Additional experiments are briefly described, but fully reported in the Appendix.

**Questions:**

* Concerning Memory-directed Active Forgetting via Machine Unlearning: the fact that you point to the neuroscience literature indicating that humans tend to forget dissonant facts, thus inspiring your design of a forgetting mechanism is in contrast with other literature [1], which suggests that instead humans retain both obsolete and new knowledge. In other words, it seems that the human brain appends, rather than replaces, obsolete knowledge. Do you think “append only updates” are feasible within an LLM?

[1] Vincent Van Veen, Marie K Krug, Jonathan W Schooler, and Cameron S Carter. Neural activity
predicts attitude change in cognitive dissonance. Nature neuroscience, 12(11):1469–1474, 2009.

* Again, in Sec. 2.3, once you establish the key ideas (which are extremely difficult to follow in technical details, as the mathematical notation requires some polishing and disambiguation), you move on to prove generalization bounds. Is this the right way to assess the benefits of active forgetting? Isn’t there a better way to measure the effectiveness of the machine unlearning method you propose?

* In Sec. 2.4., what does $\delta_{e_1,e_2}$ mean? Is this a single parameter update related to the ingestion of two consecutive edits?

* In Sec. 2.4, can you double check expression (9)? Which independence assumptions did you use? Can you explain the relation between equation (6), $p(e_2|\delta)$?

* The whole derivation for expression (15) is also relying on extremely strong assumptions: can you elaborate on their validity? For example, why do you need to assume that the optimal weight update is zero? Is this to have a Gaussian approximation with zero mean? Also, from equation (14) to (15), shouldn’t we use the FIM inverse?

* In equation (16), concerning the "MAF" loss, how do you determine what you should forget? Is this $o_1$ and $o_2$ to be replaced with the new objects? How do you locate the weights "holding" those objects?

* In Section 2.5, I am not sure to fully grasp the relation between $\delta$ and $\Delta$. My understanding is that $\Delta$ is not a matrix of learnable parameters.

* In appendix E.5, Table 2, results indicate the role of each proposed module. Can you comment on why the FIM module does not seem to play a major role, whereas the LEM module is the main responsible for improved performance, even if ER is active.

**Ethical Concerns:**

["NO or VERY MINOR ethics concerns only"]

**Final Justification:**

In light of the rebuttal, authors addressed my main concerns. I have also checked other reviews and rebuttal, and I think the authors did a good job in supporting their paper. For these reasons, I decided to considerable raise my score.

**Limitations:**

Yes, in the appendix.

**Paper Formatting Concerns:**

No concerns noticed.

**Quality:**

2

**Strengths And Weaknesses:**

- Strengths

  * Continual learning is a very timely and important topic, and I greatly appreciate the authors approach of rooting their ideas in the neuroscience literature.

  * The proposed methods, despite their complexity (e.g., machine unlearning is a field per-se, and research is on-going to improve on currently available methods), are nicely combined into a unified framework, which appears to be assessed fairly and compared to several approaches from the state of the art

  * The appendix contains several additional experimental results (e.g., an ablation study to assess the benefits of individual components of the proposed method) which confer additional solidity to the claims by the authors

- Weaknesses

  * The mathematical rigor of this work should be improved. I had a really hard time following the mathematica details and notation used in this work, as it is inconsistent, and hinders a proper understanding of the details of the proposed methods. The general idea is quite clear, but the notation needs some polishing. Also, please double check some equations, as discussed in the questions below.

  * The overall framework presented by the authors is fairly complex, and requires computation of the (inverse of) Fisher information matrix, as well as methods to determine and locate ``objects’’ to forget, update matrices and so on. I could find some discussions about complexity in appendix E.6, but I am not sure the sequential edit complexity of $n$ facts is really linear.

  * The distinction between the LEM and the ER modules is not sufficiently clear to me. This is evaluated in the ablation study in appendix E.5, and they are considered as separate entities, whereas in Sec. 2.5 they are presented as a unified mechanism.

  * Despite impressive results from the experimental evaluation of the proposed method, I am not convinced by the experimental protocol. The key ideas developed in this work are individually discussed in appendix E.5, but I think an extended version of the ablation should include more insights on the role of individual components. For example, the FIM module does not seem to play a major role, whereas the LEM + ER modules are the main responsible for improved performance.

---

> ### Author Rebuttal · Authors · 2025-07-31
>
> Dear reviewer jUcr:
>
> Thanks for your valuable feedback and constructive suggestions. Follow your advice, we have supplemented the relevant experiments and provided explanations, which have basically solved all the problems. We would greatly appreciate it if you could reconsider your final evaluation of our work.
>
> ***
>
> # **Weaknesses:**
>
> ## W1:polish mathematical details and notation
>
> In the next version, we will rename some notation and polish the descriptions of math details. We also build a mapping of notation and its definition to appendix A.
>
> We clarify some notations and will rename them:
> 　$\delta$:incremental update parameter of edited layers' output $v$ (Eq. 2).
> 　$\Delta$:incremental update matrix for modifying the edited parameters (from closed-form solution in Theorem 2).
> 　$\delta^*_{e_1,e_2}$:optimal $\delta$ when editing $e_2$ while jointly considering both $e_1$ and $e_2$.
>
> Sorry for the difficulty of reading math details, as our work involves many math details due to its substantial theoretical contributions.
>
> ## W2:computation complexity
>
> ### W2.1:framework is complex
>
> The task of LLM sequential editing is new and challenging, thus we heuristically explore some components. Each part is essential for achieving that tasks (varified in ablation studies), while some parts (Sec. 2.3 and 2.4) can theoretically ensure the performance.
>
> ### W2.2: complexity of our model: is it really linear?
>
> Yes, the time complexity is **really linear $O(n)$** as in appendix E.6.
>
> We only need to compute the Fisher information matrix (FIM) itself (inverse FIM are not used in actual loss computation).
>
> Computing about FIM requires gradients, which obtained directly during $\delta$'s backpropagation, allowing to extract d-dim diagonal Fisher values with $O(n*d)$ complexity, which is equivalent to $O(n)$ since $d$ is a constant (e.g. 4096). Using active forgetting in loss only needs $O(1)$, which modifies loss (Eq.6) without matrix computations.
>
> Our additional experiments **reports running time as the sample increasing**. It shows **time grows linearly** as the sample number (edit counts) scaling and our model's run time is less than our baseline, MEMIT_full.
>
> | #samples | HSE(Ours) | w/o FIM | FIM| MEMIT_full |w/o Active Forget |
> |-|-|-|-|-|-|
> |10|1min|1min|0.05min |3min|1min|
> |100|10.5min| 10min|0.5min|65min|11min|
> |1000|101min| 97min|4min|850min|100min|
> |3000|251min| 242min|9min| / |252min|
>
> ## W3:difference:LEM vs ER
>
> **LEM is an advanced version of ER.** Our proposed LEM is parameter-wise replay, ER is sample-wise replay (requires additional storage for samples)[D. Rolnick et al.NIPS'2019]. LEM's novelty is upgrading from sample-wise to parameter-wise, also verified in ablation study (Tab. 2).
>
> ## W4:More insights about ablation study (FIM is not so important?)
>
> Each component is necessary. **LEM play a fundamental role while FIM/AF/ER can only performs well based on LEM.**
>
> - LEM(lines 870-873) fixes model collapse and catastrophic forgetting (CF) by controlling parameter deviation (Corollary 2). Note Fig.9 shows parameter drift and CF are serious problems, so LEM is fundamental compard to FIM/AF/ER.
>
> - FIM design(lines 855-863) measures the degree of retaining prior knowledge.  Fig. 8 shows that the FIM can effectively protect critical parameter updates for edited knowledge while preventing interference between different knowledge updates.
>
> - AF module (lines 842-853) considers whether memorizes or forgets knowledge. Corollary 1 and Ablation show AF is most significant to generalization.
>
> ***
>
> # **Questions:**
>
> ## Q1:forgetting mechanism: "append only updates" are feasible for LLM?
>
> Yes, append-only updates are theoretically feasible for LLMs. While simply appending knowledge is possible, doing so effectively requires addressing critical challenges such as knowledge interference.
>
> Retaining both old and new knowledge is indeed the ideal way. To perform well based on that way, human brains employ synaptic modulation[Michael et al. Science'2004] mechanisms to achieve effectively "functional forgetting" (selecting suitable knowledge while using it instead of relying on pure append-only approaches).
>
> Similarly, our model also retain some old knowledge instead of totally removing.
>
> Therefore, we believe that **the key to append only updates is to enable the model to use knowledge correctly**, and our approach is precisely striving in this direction.
>
> ### Q2.1:Is the generalization bound suitable to assess active forgetting (AF)？
>
> Yes, as Corollary 1, the generalization bound of active forgetting-integrated MAF loss improves over traditional cross-entropy bound, **providing theoretical evidence that** the introduction of AF leads to lower errors with the goal of addressing the generalization issues regarding editing knowledge.
>
> ### Q2.2:measure effectiveness of machine unlearning?
>
> Yes, we have verified effectiveness of machine unlearning in ablation study (Fig.7，line 867-878).
>
> ## Q3:
>
> ### Q3.1:$\delta_{e_1,e_2}^\*$'s meaning?
>
> It represents incremental parameter for $v$(Eq.2) that **simultaneously satisfies both edit $e_1$ (already edited) and edit $e_2$ (to be edited)** via MAP estimation and Bayes theorem as Eq.9.
>
> ### Q3.2 Is this a single parameter update related to two consecutive edits?
>
> Yes. $\delta_{e_1,e_2}^\*$ handles the case where edit $e_1$ has been edited and we need to update parameters for edit $e_2$ considering $e_1$. We will clarify it in the final version.
>
> ## Q4:
>
> ### Q4.1:double check expression 9?
>
> OK, we confirmed Eq.9 is correct.
>
> ### Q4.2:In Sec4.2, which independence assumption did you use?
>
> The independence assumptions stem from MAP estimation principles, where \delta follows a known prior distribution P(δ) rather than being a fixed constant. This is explicitly incorporated in Theorem 1's derivation.
>
> ### Q4.3:explain relation between equation (6) and $p(e_2|\delta)$ ?
>
> They are numerically opposite. The term log $p(e_2|\delta)$ represents $log$ maximum likelihood estimation for the edit data $e_2$, which is the negative form of $L_{MAF}$ in Eq.(6) modeled via MLE(Eq.5) too.
>
> ## Q5
>
> ### Q5.1: Eq.15 relies on strong assumptions?
>
> Actually, the related full assumptions are not so strong:
>
> 1.Eq.9-11 require assumptions of MAP estimation and Bayes rules.
>
> 2.Eq.12-13 relies on Laplace approximation's assumptions.
>
> 3.Eq.14-15's assumption is in Q5.2
>
> The deducetion of Eq.15: When we edit $e_n$, log $p(\delta|e_{<n})$ = $log p(e_{n-1}|\delta)$ +...+ $log p(e_2|\delta)+ log p(\delta|e_1)$ (Bayes rule).
> We assume optimal parameters for $e_i(i<n)$, allowing Taylor expansion of log $p(e_i|\delta)$ as negative loss: constant term and first derivative ≈ 0, and second derivative ≈ $1/2\delta^T(\lambda_iF_{e_i})\delta$. Thus yield $\log(\delta|e_1,e_2,...,e_{n-1}) =  1/2\delta^T \sum_{i=1}^{n-1} (\lambda_iF_{e_i})\delta$.
>
>
> ### Q5.2: For example, why assuming optimal weight updates is zero?
>
> 1.This assumption has been validated in many continual learning studies [1,2,3].The assumption that optimal weight updates $\delta_{e_i}^{\*}$=0 in $(i+1)\_{th}$ is based on the premise: **no incremental $\delta$ is required to produce optimal representations for $e_i$ at $(i+1)\_{th}$ edit**. This logically leads to $\delta_{e_i}^*=0$ in $(i+1)_{th}$ edit.
>
> 2.In practice, since our trainable parameter $\delta$ is small (Llama3: 4096 dims), it can be easily optimized to a local optimum. As shown in our ablation study (Table 2), the FIM module remains effective under this assumption."
>
> [1]Overcoming catastrophic forgetting in neural networks, 2017.
>
> [2]Note on the quadratic penalties in elastic weight consolidation, 2018.
>
> [3]Elastic weight consolidation (ewc): Nuts and bolts, 2021.
>
> ### Q5.3: Does it have a Gaussian approximation with zero mean?
>
> No, in Appendix C.2's proof, we did not assume a zero-mean Gaussian distribution. As explained in Q5.2, $\delta_{e_i}^\*=0$ for $i+1_{th}$ edit. When substituting $\delta_{e_i}^*=0$ into Eq.35, we naturally derive Eq.14.
>
> ### Q5.4 Eq.14 and 15, shouldn't we use FIM inverse?
>
> Eqs. 14-15 do not use FIM inverse. **The FIM emerges naturally from our posterior modeling (Eqs. 33-34), where Taylor expansion links the second-order derivative to the FIM (yielding Eq. 14 and 15)**. The FIM inverse corresponds to covariance (Eq. 30), which cannot directly be used in Eqs. 14-15.
>
> ## Q6
>
> ### Q6.1:In Eq.16 (MAF loss), how to determine what to forget?
>
> Our datasets annotate knowledge to be forgotten and our editing targets, explained in** lines 669-670**.
>
> In practical application, users can assign what to forget as their demands, e.g. treat unsatisfactory LLMs' outputs as knowledge to be forgotten and treating desired  output as targets to be memorized.
>
> ### Q6.2:Is this ($o_1$ and $o_2$) to be replaced with the new objects?
>
> Yes, we designate objects ($o_1$ and $o_2$) as knowledge to be forgotten, replacing them with the target knowledge we intend to memorize.
>
> ### Q6.3:How do you locate weights "holding" those objects？
>
> As knowledge localization studies[Meng et al NIPS'2022], our localization approach is to recognize that **the second MLP linear layer in early transformer stores substantial factual knowledge(lines 107-108 and Appendix E.2)**. Notably, even knowledge targeted for forgetting was originally stored in this specific location prior to the forgetting operation.
>
> ## Q7:Sec. 2.5, relation between $\delta$ and $\Delta$? $\Delta$ is not a matrix of learnable parameters?
>
> Your understanding is correct . $\Delta$ is not a matrix of learnable parameters.
>
> $\delta$ is the incremental update to the output vector $v$, obtained via training optimization of Eq.16.
> $\Delta$ is the incremental parameter for updating the model's edit parameters, computed via $\delta$ in Eq.18.
>
> ## Q8: Why FIM does not seem to play a major role? why  LEM seems important?
>
> See the responce to W4

---

### Official Review · Reviewer_m5tL · 2025-06-29

**Clarity:** 4
**Significance:** 3
**Originality:** 3
**Rating:** 5
**Confidence:** 5

**Summary:**

This paper proposes a biologically inspired framework for continual reinforcement learning that mimics hippocampal memory mechanisms. The authors introduce a Hippocampal-like Sequential Replay (HSR) approach with three key components: (1) an active forgetting module to selectively erase outdated knowledge (analogous to hippocampal forgetting), (2) a Fisher information-guided update strategy to separate domain-specific knowledge and prevent interference (similar to Elastic Weight Consolidation using Fisher importance), and (3) a replay mechanism that consolidates long-term memories by replaying past experiences or parameter updates (inspired by hippocampal sequential replay during rest). The paper provides theoretical analysis showing improved stability (bounded parameter drift and convergence guarantees) and smaller generalization error bounds for the learning process. Empirically, the method is evaluated on sequential RL tasks, demonstrating improved retention of old tasks and integration of new tasks, with higher overall performance than baselines that either lack replay or use standard experience replay. In summary, HSR aims to balance acquiring new knowledge with mitigating catastrophic forgetting by incorporating neuroscience-inspired sequential replay into continual RL.

**Questions:**

1. How does your sequential replay compare quantitatively to standard experience replay? Clarifying this would show the added value of the hippocampal-inspired design.

2. What assumptions underlie your theoretical results? Do the bounds apply to deep networks or mainly simplified cases? And how well do they match empirical trends?

3. How would HSR handle more complex tasks or longer task sequences? Are there expected bottlenecks with memory use or Fisher computation as task complexity increases?

4. Why is the parametric memory more effective than ER? Does it prioritize important updates? Also, how sensitive is HSR to hyperparameters like forgetting rate or Fisher strength?

5. Have you observed replay patterns similar to those in the hippocampus (e.g., forward/reverse replay, reward-focused)? Even anecdotal observations would strengthen the biological analogy.

**Ethical Concerns:**

["NO or VERY MINOR ethics concerns only"]

**Limitations:**

Partially discussed. Scalability, computational cost, and applicability to more complex domains remain unclear.

**Paper Formatting Concerns:**

None significant.

**Quality:**

4

**Strengths And Weaknesses:**

Quality: The paper is technically solid with a comprehensive approach that includes both theoretical guarantees and extensive experiments. The theoretical contribution (lemmas and corollaries) offers insights into why the method should remain stable (e.g. an upper bound on parameter norm growth and improved convergence). The empirical evaluation spans multiple sequential learning benchmarks, showing that HSR significantly outperforms existing continual learning baselines in preventing performance degradation. However, there are some quality concerns: the experimental section, while thorough, focuses on relatively limited domains/tasks (e.g. simpler or well-structured environments). It is not fully clear how the approach scales to more complex or high-dimensional RL tasks (the paper only briefly mentions results on certain benchmarks). Some important baseline comparisons (e.g. directly against standard experience replay methods or other regularization-based CL methods) are present, but a deeper analysis isolating each component’s contribution is confined to an ablation study. Overall, the execution is solid, but the scope of experiments could be broader, and a few important details (like hyperparameter settings and computational costs) are left to the appendix, making it harder to assess reproducibility at a glance.

Clarity:
Strengths:
The paper is generally well-organized and clearly written. The motivation for each component is explained via the hippocampal analogy, and the authors include helpful diagrams/figures and an ablation table to illustrate the contribution of each module (e.g. showing the benefit of the long-term memory replay vs. a standard experience replay buffer). The methodology is described with sufficient mathematical detail, and important claims are theoretically justified.

Weakness:
One minor weakness in clarity is that the method involves many moving parts and new terminology (AF, Fisher update, LTM replay, etc.), which can be heavy for readers – some sections (especially the theoretical derivations) are dense. The connection between the biological inspiration and the algorithm is explained, but could be more intuitively illustrated (e.g. a schematic of how information flows through the modules would help). In summary, clarity is good, but the paper could be made more accessible by simplifying certain explanations and highlighting the main intuition behind the approach.

Significance: Continual learning in reinforcement learning is an important problem, and a method to mitigate catastrophic forgetting has practical significance. The idea of using hippocampal-like sequential replay is intriguing and could inspire further research bridging neuroscience and AI. The proposed HSR framework shows tangible improvements on the tested tasks – for instance, it maintains high performance on earlier tasks while learning new ones, unlike naive fine-tuning that quickly forgets. That said, the significance is somewhat limited by the evaluation scope. The tasks and settings presented, while demonstrating the method’s potential, do not fully convince that HSR would scale to very complex real-world continual RL scenarios (e.g. lifelong learning in rich 3D environments or continuous control with high-dimensional observations). The improvements, though consistent, appear incremental over strong baselines rather than revolutionary. Additionally, some components of HSR (such as Fisher-guided updates or experience replay) are adaptations of known techniques, so the conceptual advance is in how they are combined. In summary, the work is a useful contribution to continual RL but may be viewed as a specialized approach; its broader impact will depend on validation in more diverse settings.

Originality: The paper combines ideas from neuroscience (hippocampal memory processes) with machine learning techniques for continual learning. This interdisciplinary approach gives the work a novel flavor. In particular, the introduction of a parametric replay mechanism for long-term memory consolidation is a creative twist – rather than storing raw past experiences only, the method replays knowledge in a more abstract form (in the LLM version of this framework, a “parameter replay” was proposed). The active forgetting module for selective unlearning is also interesting and less common in typical RL algorithms. However, many elements of the solution are built on prior work: using Fisher information to protect important weights is essentially the idea from Elastic Weight Consolidation, and experience replay for continual learning is a known baseline approach. The paper’s novelty lies in the particular combination and the biological framing, rather than wholly new algorithms. It would be fair to say the work is an incremental improvement that integrates existing techniques (regularization, replay, etc.) under a unifying bio-inspired framework. This is valuable, but the originality is moderate – the authors should clearly differentiate HSR from prior approaches (e.g. how their “sequential replay” differs from standard replay buffers or recent hippocampal-inspired models).

---

> ### Author Rebuttal · Authors · 2025-07-31
>
> Dear Reviewer m5tL:
>
> We sincerely appreciate your positive feedback and constructive suggestions. As requested, we have conducted additional experiments and carefully addressed all revision requirements. We are grateful for your insightful comments, which have significantly strengthened our manuscript.
> ***
> # Questions
> ## Q1:sequential replay compare quantitatively to standard experience replay?
>
> Compared to standard experience replay, our proposed sequential replay includes all edited data through parameterization, requiring nearly no additional memory or computation costs (see Appendix C3 for proof). Standard experience replay maintains a memory buffer storing the $M$ most recent edits (replay of all M samples during Eq.18 computation), **our approach reduces both storage and computational to just 1/M of standard experience replay**.
>
> Quantitative performance comparisons are provided in Table 2 of the manuscript
> | Edit Mode | Counterfact | | |  |  |  | ZsRE  |  |  |
> |---|--|-|--|--|---|--|--|--|-|
> |     | Eff↑ | Gen↑ | Spe↑ | Flu↑ | Consis↑ | Eff↑ | Gen↑ | Spe↑ |
> | HSE (Ours)| **99.60±0.37**  | **93.80±0.51** | **87.50±0.84** | **632.76±0.43** | **32.89±0.21** | **99.28±0.65** | **96.78±0.49** | **41.90±0.31** |
> | w/o LEM, w/ ER | 81.26±0.48 | 73.50±0.93 | 76.10±0.62 | 518.62±1.08 | 14.29±0.37 | 42.50±0.43 | 38.72±0.63 | 26.14±0.78 |
> ## Q2:
>
> ### Q2.1 What assumptions underlie your theoretical results?
>
> Our theoretical results requries the following assumptions:
>
> 1.The lemma 1 and corollary 1 specific assumptions primarily follow work [1] and are formally articulated in Lemma 1(line 160).
>
> 2.Theorem 1 incorporates both the maximum a posteriori probability assumption for parameters (line 194) and the Laplace approximation of posterior probability (line 201);
>
> 3.Theorem 2 assumes local optimum is achieved for all previous $i-1$ edits during the $i_{th}$ edit (line 609);The assumptions for Corollary 2 are explicitly stated in the paper (line 243).
>
> ### **Q2.2:Do the bounds apply to deep networks or mainly simplified cases?**
>
> Under Lemma 1 and its assumptions, our bound applies for deep neural networks optimized with Adam, where the generalization error bounds are derived based on Adam's gradient update mechanism (our approach).
>
> For the simplified cases (not a  deep neural networks optimized with Adam), the bound does not applied.
>
> [1]Lipschitzness Effect of a Loss Function on Generalization Performance of Deep Neural Networks Trained by Adam and AdamW Optimizers，2023
>
> ### **Q2.3:how well do theoretical results match empirical trends?**
>
> 1.Corollary 1 shows that our proposed active forgetting-based loss MAF achieves a superior generalization error bound compared to traditional cross-entropy (CE). This theoretical advantage is empirically validated in ablation studies in Table 2 and Fig. 7, where MAF outperforms CE in multiple metrics. **The improved generalization performance align with the theoretical superiority of the error bound under generalized scenarios**.
>
> 2.Based on Theorem 1, we constructed the loss function in Eq. 16 to protect important parameters during updates, **with ablation studies in Fig. 8 visually confirming this protection;**
>
> 3.Building upon Theorem 2 and Corollary 2, we implemented parameter replay,**with ablation results in Tab. 2 and Fig. 8 empirically validating both the effectiveness of our replay and the controlled convergence of model parameters**.
>
> In summary, we have achieved a consistent match between our theoretical analysis and empirical results.
>
> ## Q3:
>
> ### **Q3.1: How would HSE handle more complex tasks or longer task sequences?**
>
> For more complex scenarios that cannot be edited in a single operation, we propose decomposing such reasoning into smaller inference units and applying sequential editing. We have supplemented additional experiments using LLaMA3-8B-Instruct on MQUAKE-CF[2] multi-hop QA dataset, evaluating sequential editing of decomposed multi-hop facts:
> | Method | MQUAKE-CF | |
> |-|-|-|
> |  | bert-score  | rouge-L |
> | Fully-FT-L| 45.87   |12.99|
> | MEND| 67.85  | 22.48|
> | ROME| 70.10  | 21.07|
> | MEMIT| 75.31   | 22.73|
> | AlphaEdit| 69.85   | 23.04|
> | HSE (Ours)| **98.42** | **96.17** |
>
> The results demonstrate that our method remains effective for long-context multi-hop reasoning tasks when employing decomposed units sequential editing.
>
> ### **Q3.2: Bottlenecks with memory use or Fisher computation as task complexity increases?**
>
> Memory use : We only require two matrices to store the Fisher matrix and cov_sum, eliminating the need for additional storage. This design ensures that memory consumption does not escalate with increasing edit operations.
>
> Fisher computation:
> 1.Computing about FIM requires its gradients, which obtained directly during $\delta$'s backpropagation, allowing us to extract d-dim diagonal Fisher values with $O(n*d)$ complexity, which is equivalent to $O(n)$ since d is a constant (e.g. 4096)
> 2.For editing time, our additional experiments compare time consumption across different scenarios:
> | samples | HSE(Ours) | w/o FIM | FIM|
> |-|-|-|-|
> | 10 | 1min | 1min | 0.05min |
> | 100| 10.5min| 10min| 0.5min  |
> | 1000 | 101min | 97min   | 4min   |
> | 3000 | 251min | 242min  | 9min   |
>
> The results demonstrate that FIM's time consumption grows slowly with edit counts, following $O(n)$ scaling.
>
> ## Q4:
>
> ### **Q4.1:Why is the parametric memory more effective than ER?**
>
> Parametric memory (LEM) can guarantee convergence in parameter F-norm **leading to edited parameters maintain compatibility with other layers’ parameters** , while ER cannot provide this assurance.
>
> LEM preserves F-norm convergence of edited parameters, demonstrating both theoretical (Corollary 2) and empirical(Fig.9) superiority over ER.
>
> ### **Q4.2 Does it prioritize important updates?**
>
> Yes, the importance of updates in parametric memory is governed by the convergence factor $\alpha_i$ (Theorem 2). Both Equation 18 and the proof in Appendix C.3 shows that higher $\alpha_i$ values correspond to greater significance of the associated knowledge edits. We provide comprehensive discussion of various $\alpha_i$ designs in Table 1.
>
> ### **Q4.3:How sensitive is HSE to hyperparameters like forgetting rate or Fisher strength?**
>
> 1.Following your advices, regarding the memorizing and forgetting rate, we conducted additional experiments using the Llama3-8B-instruct model on two datasets.
> | memory factor| forget factor | counterfact | | | | | zsre | | |
> |-|- |-|-| -|-| -|-|-|-|
> |  | | Eff | Gen |Spe | Flu| Consis | Eff | Gen |Spe|
> | 1 | 0 | 96.20 | 90.15 |86.4 | 628.19 | 30.85 | 96.25 | 94.23 | 41.06 |
> | 0.8 | 0.2 | **99.60** | **93.80** | 87.50 | 632.76 | **32.89** | **99.28** | **96.78** | 41.90 |
> | 0.6 | 0.4 | 93.10 | 91.20 | 86.10 | **632.95** | 31.75 | 97.85 | 95.12 | 40.80 |
> | 0.4 | 0.6 | 90.82 | 89.70 | 87.80 | 629.12 | 30.92 | 96.92 | 93.45 | 41.87 |
> | 0.2 | 0.8 | 85.50 | 78.30 | **88.76** | 627.83 | 30.15 | 90.78 | 87.16 | **41.96** |
>
> The improvement in Spe suggests  a reduction in interference for irrelevant knowledge. The optimal parameter configuration with memory factor = 0.8 and forget factor = 0.2 (ours hyperparameters).
>
>
> 2.Following your advices, regarding the Fisher information matrix coefficient  $\lambda_i$, we conducted additional experiments using the Llama3-8B-instruct model on two datasets.
> | fisher | Eff | Gen | Spe | Flu | Consis | Eff | Gen | Spe |
> | -| -| -| -| -| -| -|-|-|
> | $\lambda_i$ | Eff | Gen | Spe | Flu | Consis | Eff | Gen | Spe |
> | 1e-2 | 99.58 | **94.05** | 87 | 631.8 | 32.75 | 99.25 | 96.85 | 41.60 |
> | 5e-2 | 99.6 | 93.9| 87.33 | 632.12 | 32.85 | 99.21 | 96.82 | 41.8 |
> | 1e-1 | **99.6** | 93.8 | 87.5 | **632.76** | **32.89** |**99.28**| 96.78 | 41.9 |
> | 5e-1 | 99.52 | 93.65 | 87.78 | 632.27 | 32.6 | 99.18 | 96.62 | 42.02 |
> |1| 99.48 | 93.5 | **87.95** | 631.95 | 32.05 | 99.12 | 96.55 | **42.08** |
>
> The results show that as $\lambda_i$ increases, generalization performance decreases while specificity improves, with optimal overall performance achieved at  $\lambda_i=1e-1$.
>
>
> ### **Q5:Have you observed replay patterns similar to those in the hippocampus (e.g., forward/reverse replay, reward-focused)?**
>
> This is a particularly interesting and insightful observation:
> The phenomenon resembles reverse replay mechanisms, which the paper [3] primarily attributes to the "recency effect", where stronger backward replay patterns during memory retention correlate with more pronounced recency effects. In our replay framework, **we similarly observe that more recent edits tend to yield better results**, demonstrating an analogous recency effect pattern. Our sequential editing tests with 1,000 samples (shown in the table below) demonstrate that:
> | Range| ZSRE| |  |
> |-|-|-|-|
> |  | ES| PS|NS|
> | 0-200 | 96.35 | 92.81 | 41.23 |
> | 200-400  | 99.52 | 93.52 | 41.54 |
> | 400-600 | 99.65 | 94.13 | 41.75 |
> | 600-800 | 99.7  | 94.62 | 42.01 |
> | 800-1000| **99.89** | **95.03** | **42.27** |
>
> Empirical results shows that **more recent edits consistently outperform earlier ones**, manifesting a similar recency effect. We attribute this to the inherent property of continuous $\Delta$-based parameter updates, where newer knowledge experiences less interference while progressively disrupting earlier edits.
>
>
> Regarding reward-focused mechanisms, this essentially considers whether the replay process prioritizes higher-reward memories. We posit that the replay intensity for edited knowledge primarily depends on our designed convergence factor $\alpha_i$.  **Higher $\alpha_i$ values indicate more focused replay for specific edited knowledge**. The design rationale for $\alpha_i$ (Appendix C.3) and its functional impacts (Table 1) are thoroughly documented. For reward-focused scenarios, we can adaptively assign different $\alpha_i$ values to incorporate reward considerations based on practical requirements.
>
>  [3] Fast-backward replay of sequentially memorized items in humans，2018

---

### Official Review · Reviewer_h2Sa · 2025-07-01

**Clarity:** 2
**Significance:** 3
**Originality:** 3
**Rating:** 4
**Confidence:** 2

**Summary:**

This paper presents a novel framework, Hippocampal-like Sequential Editing (HSE), for continual model editing of large language models (LLMs). Drawing inspiration from the hippocampal trisynaptic circuit, the authors design three key mechanisms: (1) memory-directed active forgetting via machine unlearning, (2) domain-specific parameter stabilization using the Fisher Information Matrix (FIM), and (3) long-term memory consolidation through parametric replay. The method shows impressive empirical results, outperforming strong baselines in sequential editing benchmarks, and maintaining or even improving generalization after up to 1,000 edits.

**Questions:**

See Weakness

**Ethical Concerns:**

["NO or VERY MINOR ethics concerns only"]

**Final Justification:**

Thank you for the rebuttal. The authors have addressed some of my concerns, and I maintain my positive score.

**Limitations:**

See Weakness

**Quality:**

3

**Strengths And Weaknesses:**

Strengths:

1. The biological inspiration is well-grounded and mapped coherently to algorithmic modules.

2. The method achieves strong empirical performance across multiple LLMs and domains, including real-world tasks like bias reduction and medical knowledge injection.

3. The use of a closed-form low-rank update and FIM regularization provides theoretical and practical benefits in terms of convergence and generalization error bounds.

Weaknesses:

1. Limited editing locality and adaptability: The method focuses editing solely on the second linear layer in early transformer FFNs. This assumes fixed knowledge localization and may not generalize well to more distributed or higher-layer semantic representations.

2. Restricted capacity for complex edits: The use of low-rank additive updates (i.e., rank-one updates per edit) limits the model’s capacity to encode complex or structurally entangled knowledge edits, such as causal reasoning or multi-hop inferences.

3. Fisher matrix overhead: While the FIM-based update regulation is theoretically sound, computing and storing FIMs—even in diagonal or approximate form—can become computationally expensive as the number of edits increases.

4. Key-value encoding assumption: The method assumes that factual edits can be well-represented in a key-value format. This restricts applicability to structured knowledge (e.g., triples or QA pairs), and limits extension to more general-purpose LLM behaviors.

5. Replay via covariance only: The replay mechanism uses covariance matrices instead of actual examples or activations. This lightweight design may accumulate approximation error over time, especially under domain shifts or noisy edits.

---

> ### Author Rebuttal · Authors · 2025-07-31
>
> Dear  Reviewer h2Sa:
>
> We sincerely appreciate your insightful and positive comments. We believe all suggested experiments and explanations have been completed and addressed. We hope to continue receiving your supportive feedback during the discussion phase.
>
> # **Weaknesses:**
>
> ## W1: Limited editing locality and adaptability:  Assume fixed knowledge localization and may not generalize well to more distributed or higher-layer semantic representations.
>
> 1.Current mainstream approaches[1,2,3,4] in knowledge editing tasks for LLMs focus on the second linear layer of early transformer FFNs. **We follow this setting based on empirical evidence from causal tracing of semantic knowledge[1,3,4] and memory storage mapping[2] that substantiates this design choice**.
>
> 2.In practical, for most existing kinds of knowledge editing [1][4], we maintain that **editing on the linear layer is sufficient**. As demonstrated in [1], perturbing early linear layers plays a crucial role in model predictions, exerting greater influence than other layers. Furthermore, [4] reveals that the output patterns of early linear layers map to human-recognizable patterns in the vocabulary space.
>
> [1]Locating and editing factual associations in gpt, 2022.
>
> [2]Transformer feed-forward layers are key-value memories, 2021.
>
> [3]Alphaedit: Null-space constrained knowledge editing for language models, 2024.
>
> [4]Mass editing memory in a transformer, 2023.
>
> ## W2: Restricted capacity for complex edits: The use of low-rank additive updates (i.e., rank-one updates per edit) limits the model’s capacity to complex or structural knowledge edits, such as causal reasoning or multi-hop inferences.
>
> We indeed use low-rank to approximate the full version of the model. Current low-rank approximation research, including LoRA [5] and its variants, has demonstrated **performance comparable to fully fine-tuning with mature implementations**. Our additional experiments shows the performance of fully fine-tuning of editing layers (Fully-FT-L), which consistently show our method's superiority (Fig.2).
>
> For more complex scenarios like causal reasoning or multi-hop inference that cannot be edited in a single operation, **we can decompose a complex editting task into multiple single-step editting task and apply sequential editing**. In such the usage, we can easily adapt existing editting methods (including our method) to those complex scenarios. Further, compared to existing editting methods, our method is skilled at editting methods, which ensures the performance under a long sequecne of editting.
>
> We have supplemented additional experiments using LLaMA-3-8B-Instruct on MQUAKE-CF[6] multi-hop QA dataset, evaluating sequential editing of decomposed multi-hop facts:
> | Method       | MQUAKE-CF             |                  |
> |--------------|-----------------------|------------------|
> |              | bert-score            | rouge-L          |
> | Fully-FT-L   | 45.87                 | 12.99            |
> | MEND         | 67.85                 | 22.48            |
> | ROME         | 70.10                 | 21.07            |
> | MEMIT        | 75.31                 | 22.73            |
> | AlphaEdit    | 69.85                 | 23.04            |
> | HSE (Ours)   | **98.42**                 | **96.17**   |
>
> The results demonstrate that our method remains effective for long-context multi-hop reasoning tasks when employing decomposed units sequential editing.
>
> We will explore knowledge editting on more complex scenarios in our future tasks.
>
> [5]LoRA: Low-Rank Adaptation of Large Language Models, 2022.
>
> [6]AKEW: Assessing Knowledge Editing in the Wild, 2024.
>
> ## W3: Fisher matrix overhead: While the FIM-based update regulation can become computationally expensive as the number of edits increases.
>
> 1.Regarding computational complexity, **our method maintains an $O(n)$ theoretical complexity**.  Computing about FIM requires its gradients, which obtained directly during $\delta$'s backpropagation, allowing us to extract d-dim diagonal Fisher values with $O(n*d)$ complexity, which is equivalent to $O(n)$ since d is a constant (e.g. 4096)
>
>
> 2.For editing time, our additional experiments compare time consumption across different scenarios:
> | samples | HSE(Ours) | w/o FIM | FIM   |
> |---------|-----------|---------|-------|
> | 10      | 1min      | 1min    | 0.05min |
> | 100     | 10.5min   | 10min   | 0.5min  |
> | 1000    | 101min    | 97min   | 4min   |
> | 3000    | 251min    | 242min  | 9min   |
>
> The results demonstrate that FIM's time consumption grows slowly with edit counts, following $O(n)$ scaling.
>
> ## W4:Key-value encoding assumption: The method assumes that factual edits is well-represented in a key-value format. It restricts applicability to structured knowledge (e.g., triples or QA pairs) and limits its extensions.
>
> 1.For knowledge editting for LLMs, most current mainstream research[1,2,3,4] employs Key-Value format to encode knowledge, which can also effectively handle triples and QA pairs.
>
> 2.For more general knowledge forms like multi-hop QA in long texts, **we can decompose a complex editting task into multiple single-step editting task and apply sequential editing**, which can be modeled as key-value pairs like normal single-step editting . Our supplementary experiments on the MQUAKE-CF dataset (mentioned in W2 response) demonstrate our method's generalization ability to complex knowledge editing scenarios.
>
> ## W5:Replay via covariance only: The replay mechanism uses covariance matrices instead of actual examples or activations. This lightweight design may accumulate approximation error over time, especially under domain shifts or noisy edits.
>
> 1.We have **theoretically demonstrated in Theorem 2** that the covariance enables full replay of all edited data while controlling error accumulation across edits. Furthermore, Corollary 2 establishes the stability guarantee by proving its convergence in parameter F-norm.
>
> 2.Furthermore, our ablation studies in Table 2 demonstrate that our method with **parameter replay with covariance summation outperforms conventional experience replay** methods (replaying actual examples).

---

### Official Review · Reviewer_S7KK · 2025-07-03

**Clarity:** 3
**Significance:** 3
**Originality:** 3
**Rating:** 5
**Confidence:** 4

**Summary:**

This paper proposes a Hippocampal-like Sequential Editing (HSE) framework, aiming to address the issues of parameter drift and model collapse in the continuous knowledge update of Large Language Models (LLMs). Inspired by the continuous memory and forgetting mechanisms of the hippocampal trisynaptic circuit, the framework designs three core mechanisms: machine unlearning of obsolete knowledge, Fisher information matrix-guided parameter updates to prevent cross-domain interference, and parameter replay to consolidate long-term editing memory. Theoretical analysis shows that it has smaller generalization error bounds, more stable convergence, and higher computational efficiency. Experimental results verify its effectiveness in balancing the acquisition of new knowledge and catastrophic forgetting, as well as maintaining or even enhancing the general capabilities of the model.

**Questions:**

See above

**Ethical Concerns:**

["NO or VERY MINOR ethics concerns only"]

**Final Justification:**

The author's reply solved the problem I raised.

**Limitations:**

yes

**Quality:**

3

**Strengths And Weaknesses:**

Strengths:

1. Inspired by the continuous memory and forgetting mechanisms of the hippocampal trisynaptic circuit, this paper proposes the HSE framework, which designs three unique core mechanisms, providing new ideas and methods for solving the problem of continuous knowledge updates in LLMs.

2. This paper conducts experiments on multiple benchmark datasets and practical application scenarios, and the results show that HSE performs excellently in balancing the acquisition of new knowledge and catastrophic forgetting, as well as maintaining or even enhancing the general capabilities of the model.

Weaknesses:

1. The HSE framework involves multiple hyperparameters (such as the Fisher information matrix coefficient λi, the convergence factor αi, etc.). Although the paper provides specific settings, it does not systematically explore the sensitivity of these hyperparameters in different scenarios and the optimal adjustment strategies, which may affect its flexibility and robustness in practical applications.

2. A related method that first forgets old knowledge and then updates new knowledge has not been discussed and compared:

[1] Ni, Shiwen, et al. "Forgetting before Learning: Utilizing Parametric Arithmetic for Knowledge Updating in Large Language Models." Proceedings of the 62nd Annual Meeting of the Association for Computational Linguistics (Volume 1: Long Papers). 2024.

---

> ### Author Rebuttal · Authors · 2025-07-31
>
> Dear reviewer S7KK:
>
> We sincerely appreciate your positive feedback and constructive suggestions. As requested, we have conducted additional experiments and carefully addressed all revision requirements. We are grateful for your insightful comments, which have significantly strengthened our manuscript.
> ***
> # Weaknesses:
> ## W1:It does not systematically explore the sensitivity of these hyperparameters(such as the Fisher information matrix coefficient $\lambda_i$, the convergence factor $\alpha_i$, etc.) in different scenarios and the optimal adjustment strategies.
>
> 1.Following your suggestions, we conduct additonal experiments to systematically explore the choise of hyperparameters. We report the results that tunes the Fisher information matrix coefficient $\lambda_i$ using the Llama3-8B-instruct model on two datasets.
> | fisher |       |      | counterfact |       |                   |                | zsRE     |         |
> |--------------|--------|---|----|-------|---------|--------------|-----------|-------|
> | $\lambda_i$ | Eff     | Gen    | Spe    | Flu | Consis | Eff     | Gen    | Spe  |
> | 1e-2     | 99.58  | **94.05** | 87    | 631.8 | 32.75    | 99.25          | 96.85    | 41.60    |
> | 5e-2     | 99.6   | 93.9  | 87.33 | 632.12          | 32.85               | 99.21          | 96.82    | 41.8     |
> | 1e-1     | **99.6** | 93.8  | 87.5  | **632.76**      | **32.89**           | **99.28**      | 96.78    | 41.9     |
> | 5e-1     | 99.52  | 93.65 | 87.78 | 632.27          | 32.6                | 99.18          | 96.62    | 42.02    |
> | 1     | 99.48  | 93.5  | **87.95** | 631.95          | 32.05              | 99.12          | 96.55    | **42.08** |
>
> The results show that **as $\lambda_i$ increases, generalization performance decreases while specificity improves**, with optimal overall performance achieved at  $\lambda_i=1e-1$ (ours).
>
> 2.For the hyperparameter $\alpha_i$ adjustment, **we have provided detailed discussion in Appendix D Table 1**, including analysis of why we selected $\alpha_i=n/(i-1)$ as our configuration.
>
> 3.We will systematically explore more hyperparameters and update the results in the next version.
>
> ## W2:A related method[1] that first forgets old knowledge and then updates new knowledge has not been discussed and compared:
> [1] Forgetting before Learning: Utilizing Parametric Arithmetic for Knowledge Updating in Large Language Models. 2024.
>
> We acknowledge that F-learning indeed highlights the importance of forgetting mechanisms in model editing, and we will cite it in our subsequent work. Below **we present the performance comparison of continuous editing scenarios on two datasets using the Llama3-8B-instruct model**.
>
> | Method               |           |        counterfact          |                  | zsre                |                  |                  |
> |----------------------|----------------------|------------------|------------------|--------------------|------------------|------------------|
> |                      | ES                   | PS               | NS               | ES                 | PS               | NS               |
> | F-learning(lora-FT)  | 84.23                | 59.47            | 69.32            | 87.62              | 83.85            | 30.54            |
> | F-learning (FT)      | 75.53                | 53.56            | 68.41            | 89.65              | 88.71            | 32.18            |
> | HSE(ours)            | **99.60**                 | **93.90**             | **87.33**           |**99.21**              | **96.82**           | **41.80**            |
>
> The results demonstrate that **our method comprehensively outperforms F-learning across all evaluation metrics**.
>
> We will regard this method as a new baseline and discuss it in the related work in the  next version.

---

> > ### Comment · Reviewer_S7KK · 2025-08-07
> >
> > The author's reply resolved the doubts I raised. I raised my score to 5. Please ensure that the content of your reply will definitely be updated in the final version.

---

> > > ### Author Response · Authors · 2025-08-07
> > >
> > > Dear Reviewer S7KK,
> > >
> > > Thank you sincerely for your time and valuable feedback during the review process. We deeply appreciate your thorough evaluation and are grateful for your decision to raise the score, which demonstrates your commitment to helping improve our work.
> > >
> > > We hereby confirm that all additional experiments and baseline comparisons mentioned in our rebuttal will be properly incorporated in the final version, with careful attention to maintaining the improvements we discussed.
> > >
> > > Once again, we truly appreciate your constructive comments that have helped strengthen our paper.
> > >
> > > Best regards,
> > >
> > > Authors of paper 15767

---

### Note · Authors · 2025-08-11

Dear AC and reviewers,

We sincerely appreciate the opportunity to submit our final remarks. We would like to offer a concise summary of our contributions and the rebuttal process.

LLMs model editing is an emerging, challenging and significant research field. **Our work addresses the critical challenges of model collapse and catastrophic forgetting in sequential model editing for LLMs.** Inspired by the biological hippocampus, we propose a novel Hippocampus-like Sequential Editing (HSE) framework that tightly integrates neurobiological mechanisms with algorithmic design **(all reviewers pointed)**. Through theoretical analysis and extensive empirical evaluation across multiple practical scenarios, we demonstrate the effectiveness of our approach **(all reviewers pointed)**. Notably, the Long-Term Editing Memory (LEM) module plays a decisive role in controlling parameter drift, which represents a significant breakthrough in the field of model editing. **We have clarified this key point during the author-reviewer discussion **(reviewers m5tL, jUcr)**, reinforcing the novelty and impact of our contribution.**

Throughout the rebuttal phase, we have thoroughly addressed all concerns raised by the reviewers, including clarifications on mathematical notations and assumptions, comprehensive ablation studies on hyperparameters, analysis of computational complexity, and detailed justifications for the necessity and insights of our core components. **We are grateful that our responses have been met with consistent positive feedback, and several reviewers have acknowledged the clarity and strength of our explanations.**

Finally, we extend our deepest gratitude to all reviewers and AC for your thoughtful, constructive, and time-intensive efforts throughout the review and discussion period. Your insightful feedback has not only strengthened our paper but also helped us refine and better articulate our contributions. We are truly honored by the care and rigor you have devoted to evaluating our work.

Thank you again for your consideration and for facilitating such a meaningful and productive review process.

Best regards,

Authors of Paper 15767

---

### Decision · Program_Chairs · 2025-09-17

**Decision:**

Accept (poster)

**Comment:**

The paper presents Hippocampal-like Sequential Editing (HSE), an approach for continual knowledge editing in LLMs which seeks to alleviate parameter drift and model collapse. Inspired by the hippocampal trisynaptic circuit, the approach comprises three components: one selectively erasing outdated knowledge, one performing parameter updates while preventing cross-domain knowledge interference, and one performing parameter replay for memory consolidation. The approach is supported by theoretical analysis and demonstrated across multiple benchmark datasets and practical application.

The AC and reviewers appreciate the well-motivated approach, backed by theoretical considerations and extensive evaluation. The authors have provided several valuable clarification (e.g. on the mathematical notation and derivations), and additional results (sensitivity, timing, etc), which are all great additions to the work and should be incorporated.

As pointed out by Reviewer S7KK, the approach presented in [1] is closely related. While the authors have provided performance comparison on counterfact and zsre with Llama3-8B-instruct, evaluation of [1] on other models and also on the various downstream tasks are missing. Given the simplicity of [1] in contrast to the proposed approach, it is important to provide a more complete picture.

Even though Alpha-Edit can be considered a special case of your replay mechanism, it is also important to include its performance in all downstream evaluations studied in the main paper (in addition to the results provided in the rebuttal for the multihop dataset MQUAKE-CF.)

For the multi-hop QA datasets, the authors consider decomposing a complex editing task into multiple single-step editing task and applying sequential editing, however this is not as trivial as it seems. See e.g. [2], which points out that generating sub-questions accurately is challenging for less powerful models and that inaccurate sub-questions will yield irrelevant fact retrieval, thereby impacting editing effectiveness.
Additionally, the premise that edits should happen at shallow layers is no longer valid in the case of multi-hop problems. Indeed for such tasks it seems that LLMs retrieve information from deeper layers (see e.g. [3]). Hence the concern from Reviewer h2Sa stands as valid for multi-hop problems.

In view of the above and the need to improve presentation and provide several additional evaluation results, major rewriting is required for the present work to be suitable for publication. We strongly encourage the authors to revise their manuscript according to the above and to incorporate the relevant materials provided in the rebuttal.

[1] Ni, Shiwen, et al. "Forgetting before learning: Utilizing parametric arithmetic for knowledge updating in large language models." arXiv preprint arXiv:2311.08011 (2023).
[2] Shi, Yucheng, et al. "Retrieval-enhanced knowledge editing for multi-hop question answering in language models." CoRR (2024)
[3] Zhang, Zhuoran, et al. "Locate-then-edit for multi-hop factual recall under knowledge editing." arXiv preprint arXiv:2410.06331 (2024).